

# Complete mitochondrial genome sequence of *Labriocimbex sinicus*, a new genus and new species of Cimbicidae (Hymenoptera) from China

Yuchen Yan[1], Gengyun Niu[2], Yaoyao Zhang[3], Qianying Ren[1], Shiyu Du[1], Bocheng Lan[1] and Meicai Wei[2]

[1] Key Laboratory of Cultivation and Protection for Non-Wood Forest Trees; Lab of Insect Systematics and Evolutionary Biology, Central South University of Forestry and Technology, Changsha, China
[2] Jiangxi Normal University, Nanchang, Jiangxi, China
[3] College of Life Sciences, Nankai University, Tianjin, China

## ABSTRACT

*Labriocimbex sinicus* Yan & Wei gen. et sp. nov. of Cimbicidae is described. The new genus is similar to *Praia* Andre and *Trichiosoma* Leach. A key to extant Holarctic genera of Cimbicinae is provided. To identify the phylogenetic placement of Cimbicidae, the mitochondrial genome of *L. sinicus* was annotated and characterized using high-throughput sequencing data. The complete mitochondrial genome of *L. sinicus* was obtained with a length of 15,405 bp (GenBank: MH136623; SRA: SRR8270383) and a typical set of 37 genes (22 tRNAs, 13 PCGs, and two rRNAs). The results demonstrated that all PCGs were initiated by ATN codon, and ended with TAA or T stop codons. The study reveals that all tRNA genes have a typical clover-leaf secondary structure, except for *trnS1*. Remarkably, the secondary structures of the *rrnS* and *rrnL* of *L. sinicus* were much different from those of *Corynis lateralis*. Phylogenetic analyses verified the monophyly and positions of the three Cimbicidae species within the superfamily Tenthredinoidea and demonstrated a relationship as (Tenthredinidae + Cimbicidae) + (Argidae + Pergidae) with strong nodal supports. Furthermore, we found that the generic relationships of Cimbicidae revealed by the phylogenetic analyses based on *COI* genes agree quite closely with the systematic arrangement of the genera based on the morphological characters. Phylogenetic tree based on two methods shows that *L. sinicus* is the sister group of *Praia* with high support values. We suggest that *Labriocimbex* belongs to the tribe Trichiosomini of Cimbicinae based on adult morphology and molecular data. Besides, we suggest to promote the subgenus *Asitrichiosoma* to be a valid genus.

Corresponding author
Meicai Wei, weim@jxnu.edu.cn

## INTRODUCTION

Hymenoptera is one of the largest insect order including more than 153,000 species which possess very diverse life strategies (*Peters et al., 2017*). Currently, complete or nearly complete mitochondrial genomes have been reported for 269 hymenopteran species (NCBI,

May 2019). The Cimbicidae is a small family of the superfamily Tenthredinoidea from the phytophagous Symphyta, with about 197 valid species and 26 genera around the world. Within China, 63 species representing 13 genera have already been recorded (*Taeger, Blank & Liston, 2010*; *Yan & Wei, 2010*; *Blank et al., 2012*; *Yan & Wei, 2013*; *Yan, Xiao & Wei, 2014*; *Yan & Wei, 2016*; *Yan et al., 2018*). The monophyly of Tenthredinoidea is supported by both morphological (*Wei & Nie, 1997*) and molecular data (*Malm & Nyman, 2015*) as well as both combined (*Ronquist et al., 2012*; *Sharkey et al., 2012*; *Klopfstein et al., 2013*). However, the relationships among core tenthredinoids are unclear. Cimbicidae was inferred as the sister to Argidae + Pergidae proposed by morphological analyses (*Wei & Nie, 1997*; *Vilhelmsen, 2001*; *Vilhelmsen, 2015*; *Vilhelmsen, 2019*). The disaccord with several recent studies may be caused by the limited dataset of Cimbicidae, by molecular or combined analyses, which have placed Cimbicidae as sister to Diprionidae (*Schulmeister, 2003*; *Schmidt & Walter, 2014*; *Isaka & Sato, 2015*; *Malm & Nyman, 2015*) or a clade including Diprionidae form a monophylum as sister to the remaining tenthredinoids (*Heraty et al., 2011*; *Ronquist et al., 2012*; *Klopfstein et al., 2013*). The monophyly of Cimbicidae has never been contested and not comprehensively tested until *Vilhelmsen (2019)*. Adult Cimbicidae are primarily characterized by their clubbed antennae, one or more of the apical antennomeres being expanded. They vary in size from small ( six mm) to very large insects (30 mm), making them the largest true sawflies known (*Vilhelmsen, 2019*). Some of the species are economically important pests causing serious defoliation of woody plants such as elm (*Ulmus*, Ulmaceae), willow (*Salix*, Salicaceae), honeysuckle (*Lonicera*, Caprifoliaceae) and cherry (*Prunus*, Rosaceae) (*Gauld & Bolton, 1988*). *Malaise (1934)* established the classification of Cimbicidae: subfamily, tribe, subtribe and genus. *Benson (1938)* carried out a comprehensive study of sawflies, especially the members of Cimbicidae, which was further determined by the classification status of Cimbicidae. It included four subfamilies: Abiinae, Cimbicinae, Pachylostictinae and Corynidinae. The Cimbicinae is the most diverse subfamily, it was divided into Cimbicini and Trichiosomini (*Abe & Smith, 1991*). The monophyly of Cimbicini was not supported by a following cladistic analyses with sufficient representation of cimbicid taxa of China (*Deng, 2000*) and a cladistic analyses with most representatives of cimbicid taxa of world (*Vilhelmsen, 2019*). The monophyly of Trichiosomini was supported by a cladistic analyses with sufficient representation of cimbicid taxa of China (*Deng, 2000*). The clade Abiinae + Cimbicinae received strong support in *Vilhelmsen (2019)*.

*Vilhelmsen (2019)* placed *Labriocimbex* into Cimbicinae but the name is a nomina nudum. So far, mitochondrial genome of two species in the family, *Trichiosoma anthracinum* (KT921411) and *Corynis lateralis* (KY063728) have been reported (*Song et al., 2016*; *Doğan & Korkmaz, 2017*). Here, we reported one complete mitochondrial genome of *Labriocimbex sinicus*. We also compared it with the previously reported mitochondrial genomes of *T. anthracinum* and *C. lateralis* for better understanding of the mitochondrial genome characteristics of the Cimbicidae. Finally, we have performed phylogenetic analyses to confirm the sister group relation of *Labriocimbex* and to clarify the systematic position of Cimbicidae within Symphyta.

## MATERIALS & METHODS

### Description of new species

Specimens were examined with a Leica S8APO dissection microscope. Adult images were taken with a Nikon D700 digital camera and a series of images edited using Helicon Focus (HeliconSoft), while detailed images were taken with Leica Z16 APO/DFC550. A cylinder of semitransparent plastic was placed around the specimen to disperse the light, that methods follows *Vilhelmsen (2019)*. The specimen must be sufficiently relaxed in a moist chamber before dissection. Dissected ovipositor valves, gonoforcep and penis valves were permanently mounted on slides in gum Arabic and images produced and composited automatically with a Nikon Ci-L/DS-Fi3. We used Adobe Photoshop CS 6.0 for further image processing. The terminology of sawfly genitalia follows *Ross (1945)*, and that of general morphology follows *Viitasaari (2002)*. For a few terms (e.g., middle fovea and lateral fovea), we followed *Takeuchi (1952)*. Abbreviations used were: OOL = distance between the eye and outer edge of lateral ocelli; POL = distance between the mesal edges of the lateral ocelli; OCL = distance between a lateral ocellus and the occipital carina or hind margin of the head.

The holotype and some paratypes of the new species are deposited in the Asian Sawfly Collection, Nanchang, China (ASCN). The most remaining paratypes are deposited in the Insect Collection of Central South University of Forestry and Technology, Changsha, Hunan, China (CSCS). A few paratypes are kept in Lishui Academy of Forestry (LSAF).

The electronic version of this article in Portable Document Format (PDF) will represent a published work according to the International Commission on Zoological Nomenclature (ICZN), and hence the new names contained in the electronic version are effectively published under that Code from the electronic edition alone. This published work and the nomenclatural acts it contains have been registered in ZooBank, the online registration system for the ICZN. The ZooBank LSIDs (Life Science Identifiers) can be resolved and the associated information viewed through any standard web browser by appending the LSID to the prefix http://zoobank.org/. The LSID for this publication is: urn: lsid: zoobank.org: pub: EE7F5193-78B2-42CE-87C1-B3FE947CB70F. The online version of this work is archived and available from the following digital repositories: PeerJ, PubMed Central and CLOCKSS.

### DNA library construction and sequencing

Total DNA was extracted from *L. sinicus* using an E.Z.N.A.® Tissue DNA Kit (Omega, Norcross, GA) and was stored at −20 °C, in accordance with the manufacturer's instructions. Sequencing libraries with approximately 250-bp insertions were constructed using a NEXT flex™ Rapid DNA-Seq Kit (Illumina, San Diego, CA, USA) in accordance with the manufacturer's protocol. Each library was sequenced using an Illumina Hiseq 4,000 to generate 150-bp paired end reads at BGI-Shenzhen, China. The sequencing reads have been deposited in NCBI SRA database under accession number: PRJNA507477.

## Mitochondrial genome assembly

Next generation sequencing and bioinformatic analyses were performed by Shanghai Majorbio Bio-pharm Technology Co., Ltd. Reconstruction of the mitochondrial genome from Illumina reads was carried out using three different approaches to ensure the accuracy of the assemblies: SOAPdenovo v2.0 (*Luo et al., 2012*), MITObim v1.8 (*Hahn, Bachmann & Chevreux, 2013*) and NOVOPlasty v2.7.1 (*Dierckxsens, Mardulyn & Smits, 2017*). The assembled mitochondrial fragments were identified using BlastX and *T. anthracinum* (NC029733) mitochondrial genes as queries. Prediction and annotation of protein-coding, tRNA and rRNA genes were performed using DOGMA (http://dogma.ccbb.utexas.edu/) or MITOS (http://mitos.bioinf.uni-leipzig.de/index.py) with annotation from a reference mitochondrial genome. Queries were then corrected manually.

## Mitochondrial genome annotation and secondary structure prediction

All RNA genes were identified by employing the online MITOS tool (http://mitos.bioinf.uni-leipzig.de/index.py) (*Bernt et al., 2013*) with the invertebrate mitochondrial genetic code. The initiation and termination codons of PCGs were determined using Geneious v11.0.3 (http://www.geneious.com) with reference sequences from other symphytan species with subsequent manual adjustment. The A + T content of nucleotide sequences and relative synonymous codon usage (RSCU) were calculated using MEGA v7.0 (*Kumar, Stecher & Tamura, 2016*). Strand asymmetry was calculated using the formulae (*Perna & Kocher, 1995*): GC–skew = $(G - C)/(G + C)$ and AT–skew = $(A - T)/(A + T)$, for the strand encoding the majority of the PCGs.

The secondary structures of the *rrnS* and *rrnL* were partitioned into four areas and six areas, respectively. The secondary structures of rRNAs were inferred using alignment to models predicted for *T. anthracinum*. First, the primary sequence and the secondary structure of this species were aligned in MARNA (*Siebert & Backofen, 2005*) to identify a consensus sequence as well as a consensus structure in the output files. Secondly, the secondary structures of the *rrnS* and *rrnL* in *L. sinicus* were predicted by specific structure models in SSU-ALIGN (*Nawrocki, 2009*). Finally, the structures were artificially transformed to their relative secondary structure with micro changes.

The predicted secondary structures of RNAs were drawn using VARNA v3-93 (*Darty, Denise & Ponty, 2009*) and RNAviz 2.0.3 (*De Rijk, Wuyts & De Wachter, 2003*). Helix numbering was performed following the *Apis mellifera* rRNA secondary structure (*Gillespie et al., 2006*) including minor modifications.

## Taxon sampling

We sampled all known mitochondrial genomes representatives from the symphytan of Hymenoptera (34 species of Symphyta and two representatives of Apocrita) and four Non-hymenopteran outgroups (Mecoptera, Diptera, Megaloptera, Coleoptera) including some mitochondrial genomes downloaded from GenBank which had previously been sequenced, and the newly sequenced mitochondrial genome in this study (Table 1).

To investigate the phylogenetic relationship of *Labriocimbex* within Cimbicidae, we used 40 species (43 samples) of seven genera belonging to three subfamilies according the

**Table 1  Summary information of symphytan mitochondrial genomes used in phylogenetic analyses.**

|  | Species | Family | Accesion number | References |
|---|---|---|---|---|
| Ingroup | *Labriocimbex sinicus* | Cimbicidae | MH136623 | This study |
|  | *Corynis lateralis* | Cimbicidae | KY063728 | *Doğan & Korkmaz (2017)* |
|  | *Trichiosoma anthracinum* | Cimbicidae | KT921411 | *Song et al. (2016)* |
|  | *Megalodontes cephalotes* | Megalodontesidae | MH577058 | *Niu et al. (2019a)* |
|  | *Megalodontes spiraeae* | Megalodontesidae | MH577059 | *Niu et al. (2019a)* |
|  | *Megalodontes quinquecinctus* | Megalodontesidae | MG923502 | *Tang et al. (2019)* |
|  | *Analcellicampa xanthosoma* | Tenthredinidae | MH992752 | *Niu et al. (2019b)* |
|  | *Allantus luctifer* | Tenthredinidae | KJ713152 | *Wei, Niu & Du (2014)* |
|  | *Asiemphytus rufocephalus* | Tenthredinidae | KR703582 | *Song et al. (2016)* |
|  | *Monocellicampa pruni* | Tenthredinidae | JX566509 | *Wei, Wu & Liu (2015)* |
|  | *Tenthredo tienmushana* | Tenthredinidae | KR703581 | *Song et al. (2015)* |
|  | *Birmella discoidalisa* | Tenthredinidae | MF197548 | GY Niu, 2017, unpublished data |
|  | *Xyela sp.* | Xyelidae | MG923517 | *Tang et al. (2019)* |
|  | *Xiphydria sp.* | Xiphydriidae | MH422969 | *Ma et al. (2019)* |
|  | *Tremex columba* | Siricidae | MH422968 | *Ma et al. (2019)* |
|  | *Pamphilius sp.* | Pamphiliidae | MG923504 | *Tang et al. (2019)* |
|  | *Chinolyda flagellicornis* | Pamphiliidae | MH577057 | *Niu et al. (2019a)* |
|  | *Orussus occidentalis* | Orussidae | FJ478174 | *Dowton et al. (2009)* |
|  | *Arge similes* | Argidae | MG923484 | *Tang et al. (2019)* |
|  | *Arge bella* | Argidae | MF287761 | *Du et al. (2018)* |
|  | *Calameuta filiformis* | Cephidae | KT260167 | *Korkmaz et al. (2016)* |
|  | *Calameuta idolon* | Cephidae | KT260168 | *Korkmaz et al. (2016)* |
|  | *Cephus cinctus* | Cephidae | FJ478173 | *Dowton et al. (2009)* |
|  | *Cephus pygmeus* | Cephidae | KM377623 | *Korkmaz et al. (2015)* |
|  | *Cephus sareptanus* | Cephidae | KM377624 | *Korkmaz et al. (2015)* |
|  | *Characopygus scythicus* | Cephidae | KX907848 | *Korkmaz et al. (2018)* |
|  | *Hartigia linearis* | Cephidae | KX907843 | *Korkmaz et al. (2018)* |
|  | *Janus compressus* | Cephidae | KX907844 | *Korkmaz et al. (2018)* |
|  | *Pachycephus cruentatus* | Cephidae | KX907845 | *Korkmaz et al. (2018)* |
|  | *Pachycephus smyrnensis* | Cephidae | KX907846 | *Korkmaz et al. (2018)* |
|  | *Syrista parreyssi* | Cephidae | KX907847 | *Korkmaz et al. (2018)* |
|  | *Trachelus iudaicus* | Cephidae | KX257357 | *Korkmaz et al. (2017)* |
|  | *Trachelus tabidus* | Cephidae | KX257358 | *Korkmaz et al. (2017)* |
|  | *Perga condei* | Pergidae | AY787816 | *Castro & Dowton (2005)* |
|  | *Taeniogonalos taihorina* | Trigonalidae | NC027830 | *Wu et al. (2014)* |
|  | *Parapolybia crocea* | Vespidae | KY679828 | *Peng, Chen & Li (2017)* |
| Outgroup | *Paroster microsturtensis* | Dytiscidae | MG912997 | *Hyde et al. (2018)* |
|  | *Neopanorpa phlchra* | Panorpidae | FJ169955 | J Hua, 2016, unpublished data |
|  | *Neochauliodes parasparsus* | Corydalidae | KX821680 | *Zhao, Zhang & Zhang (2017)* |
|  | *Anopheles gambiae* | Culicidae | L20934 | *Beard, Hamm & Collins (1993)* |

classification system of *Abe & Smith (1991)*, by using partial cytochrome oxidase subunit I (*COI*) gene of mitochondrial genome. Composite of ingroup and outgroup taxon, as indicated in Table 2, were developed by sequences from different taxon either from our own sequences or those deposited in GenBank.

## DNA extraction, polymerase chain reaction (PCR) and sequencing of *COI* gene

Total genomic DNA was extracted from muscles or single leg of adult specimens stored in ethanol at −20 °C using the DNeasy Blood & Tissue Kit (Qiagen, Hilden, Germany). PCRs (50 μl) contained 25μl Taq MasterMix (CW0682M) reaction buffer, 2 μl of each primer, 2 μl DNA template and 19 μl PCR grade H2O. The PCR programme consisted of an initial denaturing step at 95 °C for one min, followed by 42 cycles of 20 s denaturing at 95 °C, 30 s annealing at 55 °C and 80 s extension at 68 °C; the last cycle was followed by a final 7 min extension step at 68 °C. The primers of COI referenced *Nyman et al. (2006)*. Purified PCR products were sequenced in both directions with the BigDye v3.1 Mix Sequencing Kit (Applied Biosystems) and an ABI 3730 XL automated sequencer (Applied Biosystems). Sequence trace files were read and edited using Sequencher v4.8 (BGI-Shenzhen cop, China).

*COI* gene sequences were checked and assembled in Geneious v11.0.3 (http://www.geneious.com), prior to submission to GenBank (accession numbers MN076590–MN076605).

## Phylogenetic analysis

We used the Maximum Likelihood (ML) and Bayesian Inference (BI) methods to construct phylogenetic trees. A total of 13 PCGs were aligned by MUSCLE in MEGA v7.0 individually, two rRNAs and *COI* gene were aligned by MAFFT (https://www.ebi.ac.uk/Tools/msa/mafft/) (*Katoh & Standley, 2013*). Then, these (13 PCGs and two rRNAs) aligned nucleotide sequences were concatenated using SequenceMatrix v1.7.8 (*Vaidya, Lohman & Meier, 2011*) and partitioned into several data blocks.

The partitioned data block file was used to infer both partition schemes and substitution models in PartitionFinder v1.1.1 (*Lanfear et al., 2012*), with "unlinked" branch lengths under the "greedy" search algorithm. The standard partitioning schemes "bic" and "aicc" were selected for BI and ML analyses, respectively. BI analyses were conducted with the GTR+I+G model and HKY+G model using MrBayes v3.2.2 (*Ronquist et al., 2012*). Four simultaneous Markov chains (three cold, one heated) were run for five million generations in two independent runs, with sampling every 1,000 generations and the first 25% of generations were discarded as burn-in.

The best partitioning scheme were chosen for 13 PCGs (Table 3). The best-fit model of nucleotide substitution and phylogenetic construction based on ML were created using the IQ-TREE web server (http://iqtree.cibiv.univie.ac.at/). The previous data block file was used as well as the original parameters. In addition, 0.1 was employed as the disturbance intensity and 1,000 as the IQ-TREE stopping rule.

The *COI* gene phylogenetic construction also based on ML and BI analyses. All related files have been uploaded to Figshare (https://doi.org/10.6084/m9.figshare.7339334.v1).

**Table 2  Specimens information of Cimbicidae and accession numbers of the GenBank sequences used in phylogenetic analyses.**

| Famil | Subfamily | Genus*Species | Accesion number | References |
|---|---|---|---|---|
| Cimbicidae (ingroup) | Cimbicinae | *Labriocimbex sinicus* | MH136623 | This study |
| | | *Labriocimbex sinicus* | MN076590 | This study |
| | | *Labriocimbex sinicus* | MN076591 | This study |
| | | *Praia taczanowskii* | KF936545 | *Malm & Nyman (2015)* |
| | | *Praia taczanowskii* | KC976900 | *Schmidt et al. (2017)* |
| | | *Cimbex americana* | EF032218 | *Schulmeister, Wheeler & Carpenter (2002)* |
| | | *Cimbex fagi* | KC972801 | *Schmidt et al. (2017)* |
| | | *Cimbex luteus* | KC973384 | *Schmidt et al. (2017)* |
| | | *Cimbex femoratus* | KC976129 | *Schmidt et al. (2017)* |
| | | *Cimbex sp.* | KF936524 | *Malm & Nyman (2015)* |
| | | *Trichiosoma anthracinum* | KT921411 | *Song et al. (2016)* |
| | | *Trichiosoma aenescens* | JX090784 | *Leppänen et al. (2012)* |
| | | *Trichiosoma triangulum* | KR895520 | *Hebert et al. (2016)* |
| | | *Trichiosoma sp.* | KR878237 | *Hebert et al. (2016)* |
| | | *Trichiosoma lucorum* | KF936518 | *Malm & Nyman (2015)* |
| | | *Trichiosoma tibiale* | KC976817 | *Schmidt et al. (2017)* |
| | | *Trichiosoma sorbi* | KJ402312 | *Liston et al. (2014)* |
| | | *Leptocimbex sp.* | KC976797 | *Schmidt et al. (2017)* |
| | | *Leptocimbex sp.* | KC976130 | *Schmidt et al. (2017)* |
| | | *Leptocimbex sp.* | KC975295 | *Schmidt et al. (2017)* |
| | | *Leptocimbex_potanini* | MN076592 | This study |
| | | *Leptocimbex_linealis* | MN076593 | This study |
| | | *Leptocimbex_afoveata* | MN076594 | This study |
| | | *Leptocimbex_lii sp.* | MN076595 | This study |
| | | *Leptocimbex sp.* | MN076596 | This study |
| | | *Leptocimbex_tuberculatus* | MN076597 | This study |
| | | *Leptocimbex sp.* | MN076598 | This study |
| | | *Leptocimbex grahami* | MN076599 | This study |
| | Abiinae | *Abia niui* | MN076604 | This study |
| | | *Abia berezowski* | MN076605 | This study |
| | | *Abia candens* | DQ302235 | *Nyman et al. (2006)* |
| | Corynidinae | *Corynis lateralis* | KY063728 | *Doğan & Korkmaz (2017)* |
| | | *Corynis andrei* | KF642787 | *Schmidt et al. (2017)* |
| | | *Corynis crassicornis* | KF936612 | *Malm & Nyman (2015)* |
| | | *Corynis mutabilis* | KF642872 | *Schmidt et al. (2017)* |
| | | *Corynis krueperi* | KF642852 | *Schmidt et al. (2017)* |
| | | *Corynis sanguinea* | KF642775 | *Schmidt et al. (2017)* |
| | | *Corynis hispanica* | KF642727 | *Schmidt et al. (2017)* |
| | | *Corynis enslini* | KF642703 | *Schmidt et al. (2017)* |
| | | *Corynis caucasica* | KF642648 | *Schmidt et al. (2017)* |
| | | *Corynis obscura* | KC976955 | *Schmidt et al. (2017)* |
| | | *Corynis atricapilla* | KC976741 | *Schmidt et al. (2017)* |
| | | *Corynis italica* | KC975057 | *Schmidt et al. (2017)* |
| Argidae (outgroup) | Arginae | *Arge similis* | MG923484 | *Tang et al. (2019)* |
| | | *Arge bella* | MF287761 | *Du et al. (2018)* |

**Table 3** **The partitions of the mitochondrial genome sequences identified by PartitionFinder.**

| Partitions | Best Models | Sites | Genes |
|---|---|---|---|
| P1 | GTR +I +G | 1,568 | *rrnS* |
| P2 | GTR +I +G | 2,032 | *rrnL* |
| P3 | GTR +I +G | 751 | *ND3*_1st, *ATP6*_1st, *CYTB*_1st |
| P4 | GTR +I +G | 1,784 | *ND3*_2nd, *ATP6*_2nd, *COX3*_2nd, *CYTB*_2nd, *COX2*_2nd, *COX1*_2nd |
| P5 | GTR +I +G | 1,849 | *ND3*_3rd, *COX2*_3rd, *ATP8*_3rd, *ATP6*_3rd, *COX1*_3rd, *COX3*_3rd, *CYTB*_3rd |
| P6 | F81 +G | 130 | *ATP8*_1st, *ATP8*_2nd |
| P7 | GTR +I +G | 522 | *COX1*_1st |
| P8 | GTR +I +G | 511 | *COX3*_1st, *COX2*_1st |
| P9 | GTR +I +G | 906 | *ND4L*_1st, *ND4*_1st, *ND1*_1st |
| P10 | GTR +I +G | 906 | *ND4*_2nd, *ND1*_2nd, *ND4L*_2nd |
| P11 | GTR +G | 906 | *ND1*_3rd, *ND4*_3rd, *ND4L*_3rd |
| P12 | GTR +I +G | 599 | *ND6*_1st, *ND2*_1st |
| P13 | GTR +I +G | 599 | *ND6*_2nd, *ND2*_2nd |
| P14 | GTR +G | 599 | *ND6*_3rd, *ND2*_3rd |
| P15 | GTR +G | 604 | *ND5*_1st |
| P16 | GTR +G | 604 | *ND5*_2nd |
| P17 | HKY +G | 603 | *ND5*_3rd |

# RESULTS AND DISCUSSION

## Description

*Labriocimbex* Yan & Wei, gen. nov.
urn: lsid: zoobank. org: act: 29EB6C0E-881D-46E2-AEF0-3BDF5992EC37

**Type species:** *Labriocimbex sinicus* Yan & Wei, sp. nov.
Description. Body middle to large-sized; black, without metallic luster (Fig. 1); head and thorax with dense and long yellowish brown hairs; clypeus distinctly broader than distance between lower margin of eyes, anterior margin with arcuate notch, furrow between clypeus and supraclypeal area deep (Fig. 2A); base of labrum much broader than apex, lateral margin of labrum distinctly narrowed upward (Fig. 2A); mandibles elongate, with three teeth in total, basal one truncate at apex (Figs. 3A, 3B); maxillary palp with 6 palpomeres, apex 1–2 combined distinctly shorter than palpomere 4; labial palp with 4 palpomeres, short (Fig. 2G); malar space 2.3 times the diameter of lateral ocellus, about as long as scape and pedicel combined; eyes moderately large, inner margins parallel, distance between eyes slightly longer than longest axis of eye (Figs. 2A, 2B); lateral part of head distinctly dilated behind eyes in lateral view (Fig. 2B) and dorsal view (Fig. 2D); postocellar area with median and lateral furrows distinct, frontal carina indistinct (Fig. 2D). Antenna longer than breadth of head, club of antenna strongly enlarged with obscure annular suture, with 5 antennomeres before club, antennomere 3 slender and distinctly longer

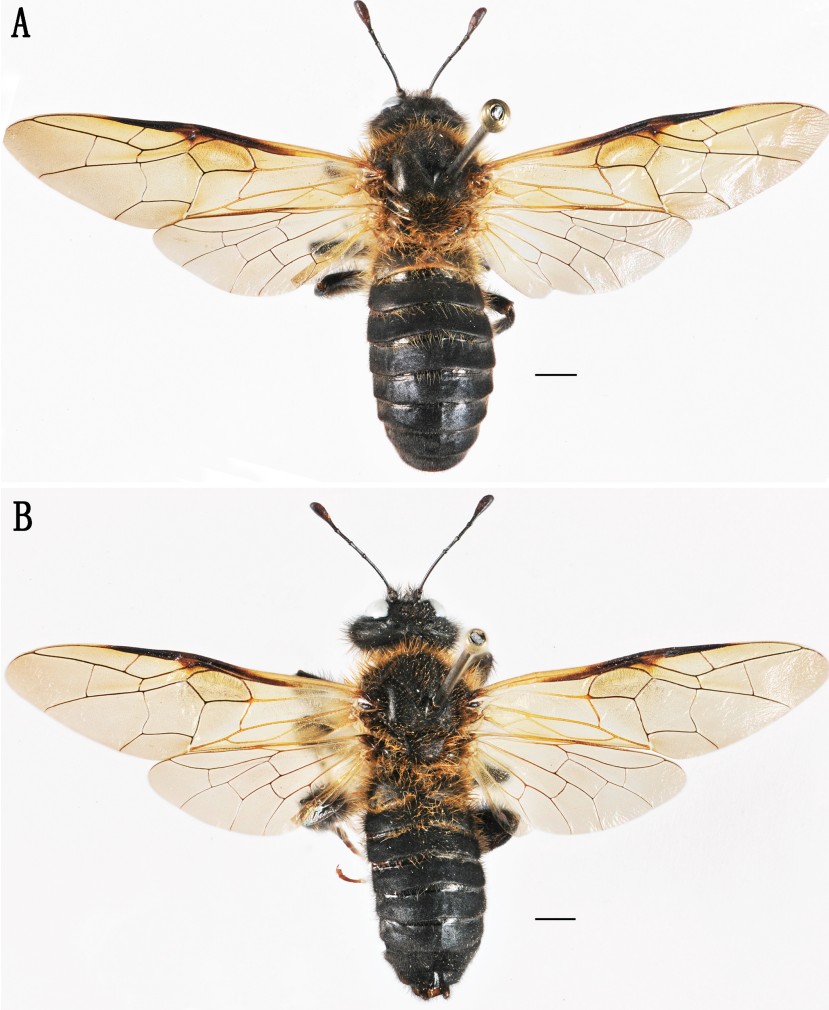

**Figure 1** *Labriocimbex sinicus* **Yan & Wei sp. nov.** (A) Adult female (holotype), dorsal view; (B) Adult male (paratype), dorsal view. Scale bar = 2 mm.

than antennomeres 4 and 5 combined (Fig. 2H). Propleuron and sternum merged; median suture of mesonotum shallow, notaulus distinct; mesoscutellum flat, anterior margin subtruncate, posterior margin roundly triangular (Fig. 2E); distance between inner margin of cenchri 3.3 times the longest axis of a cenchrus, distance between outer margin of cenchri longer than breadth of mesoscutellum (Fig. 2E). Coxae and femur of leg with long hairs; ventral side of middle and hind femur without tooth near apex (Figs. 2F, 3C), hind coxae close to each other; inner spur of hind tibia as long as apical breadth of tibia, apex blunt and membranous (Fig. 3J), about 0.4 times length of metabasitarsus; metabasitarsus slightly shorter than tarsomeres 2 and 3 combined, base of hind tibia narrower than apex (Fig. 2F); 1st and 2nd tarsal pulvilli long, nearly contiguous (Fig. 2F); claw simple, roundly bent (Fig. 3K).Fore wing with crossvein 2r present, base of vein Rs absent (Fig. 1A); vein 2r-m and 2m-cu almost interstitial, pterostigma long and narrow; anal cell strongly narrowed

in basal 1/3 with a short anal crossvein, apical anal cell about 2 times the length of basal anal cell; cell Rs and M closed in hind wing, apex of anal cell quadrate, petiole of anal cell longer than length of vein cu-a, jugum region only with 1 longitudinal vein, without crossvein (Fig. 1A). Sternites and basal abdominal terga with long hairs, posterior margin of abdominal tergum 1 shallowly incised, without middle carina and lateral carina (Figs. 2E, 2F). Genital plate of female developed with wide incision and arcuate in middle (Fig. 3L); apical ovipositor sheath short and roundish in lateral view (Fig. 3D), tapering toward apes in dorsal view (Fig. 3F); apex of lancet and lance curved upwards (Figs. 3M, 3N), each annulus with 1 pore, serrulae sub-truncate at apex, lateral teeth sharp (Fig. 3G). Each sternite of male incised in middle, both sides roundish (Fig. 3E); penis valve broad, with apical lobe bulge, ventral hook small, lateral ridge distinct (Fig. 3H); harpe small, longer than broad (Fig. 3I).

**Etymology.** The generic name is composed of "*labrio-*" and "*-cimbex*", emphasizing the shape of labrum differs from other genera of the family. Gender masculine.

**Distribution.** China.

**Host plant**: *Prunus pseudocerasus* of Rosaceae (Female adult were observed laying eggs on it).

**Remarks.**

The name *Labriocimbex* was mentioned in two papers before (*Li & Wu, 2010*; *Vilhelmsen, 2019*). The name is a nomina nuda in these papers as it was not accompanied by a proper description and type designation. The name was originally proposed by the senior author of this paper (MW) for the genus hear described as new to science. The present paper constitutes the proper establishment of the name.

This new genus is similar to *Praia* Andre and *Trichiosoma* Leach. It is similar to *Praia* having the head and thorax with dense and long hairs; the antenna with 5 antennomeres before the club; the coxae and femur of leg with long hairs; the ventral side of hind femur without a denticle near apex; the base of hind tibia narrower than apex; the 1st and 2nd tarsal pulvilli long. It differs from *Praia* by having the triangular labrum, the base of labrum much wider than its apex and the basal breadth about half the breadth of clypeus; the inner margins of eyes parallel; the anal cell strongly narrowed in basal 1/3 with a short crossvein and apical anal cell about 2 times the length of basal anal cell; the abdomen all black, without transverse band. It differs from *Trichiosoma* by the ventral side of the hind femur without a subapical denticle; the 1st and 2nd tarsal pulvilli in the male very long and nearly contacting to each other, and the different pattern of the male genitalia.

Trichiosomini includes 3 genera: *Trichiosoma*, *Pseudoclavellaria* and *Leptocimbex* (*Abe & Smith, 1991*; *Deng, 2000*). Most of the characteristics of the new genus suggest placing it in the tribe Trichiosomini. The most important characteristics include: the labrum large with the basal breadth about half the breadth of clypeus, the jugum region in hind wing without crossvein, the clypeus very short and much broader than lower distance between eyes and not merging with supraclypeal area. See the key below for the relationship of the new genus and other genera of the family.

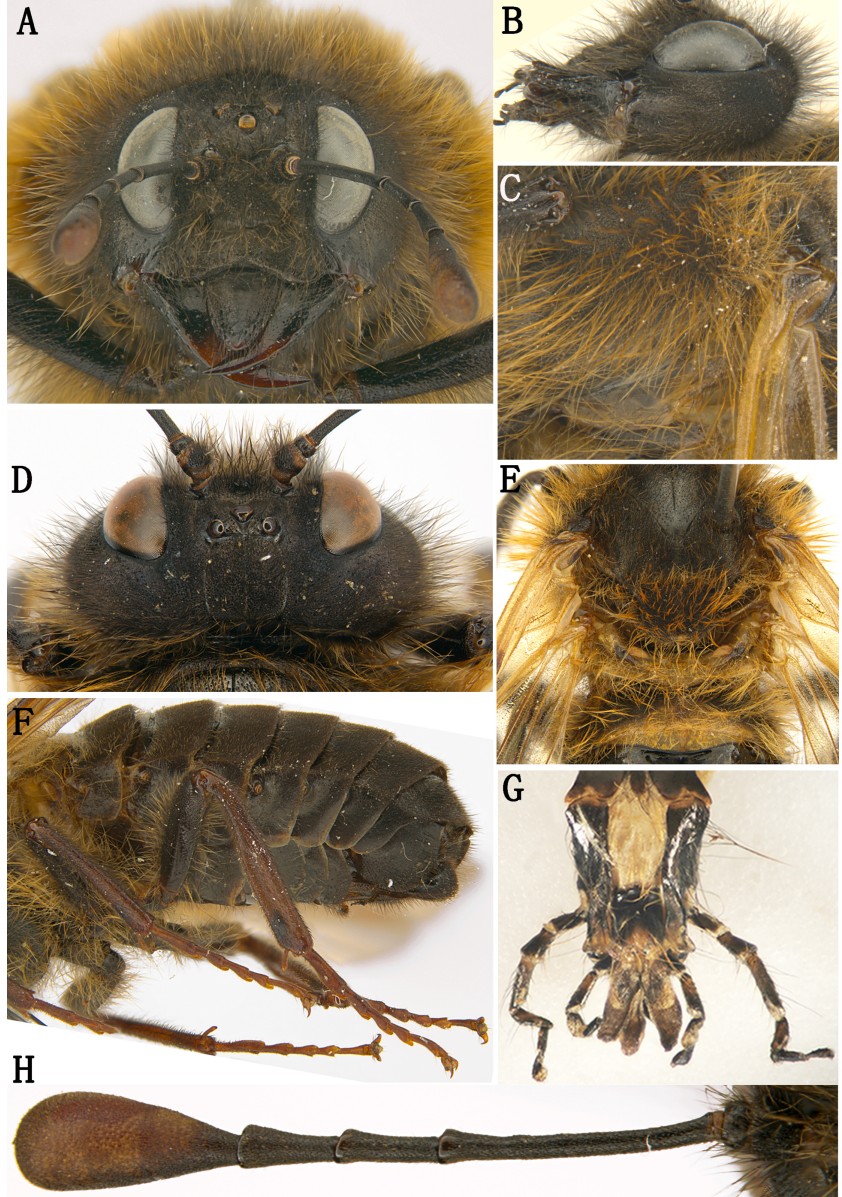

**Figure 2** *Labriocimbex sinicus* **Yan & Wei, gen. et sp. nov.** (A) Head of female, front view; (B) Head of female, lateral view; (C) Mesopleuron of female, lateral view; (D) Head of female, dorsal view; (E) Metanotum and base of abdomen; (F) Abdomen, lateral view; (G) Palpus; (H) Antenna of female.

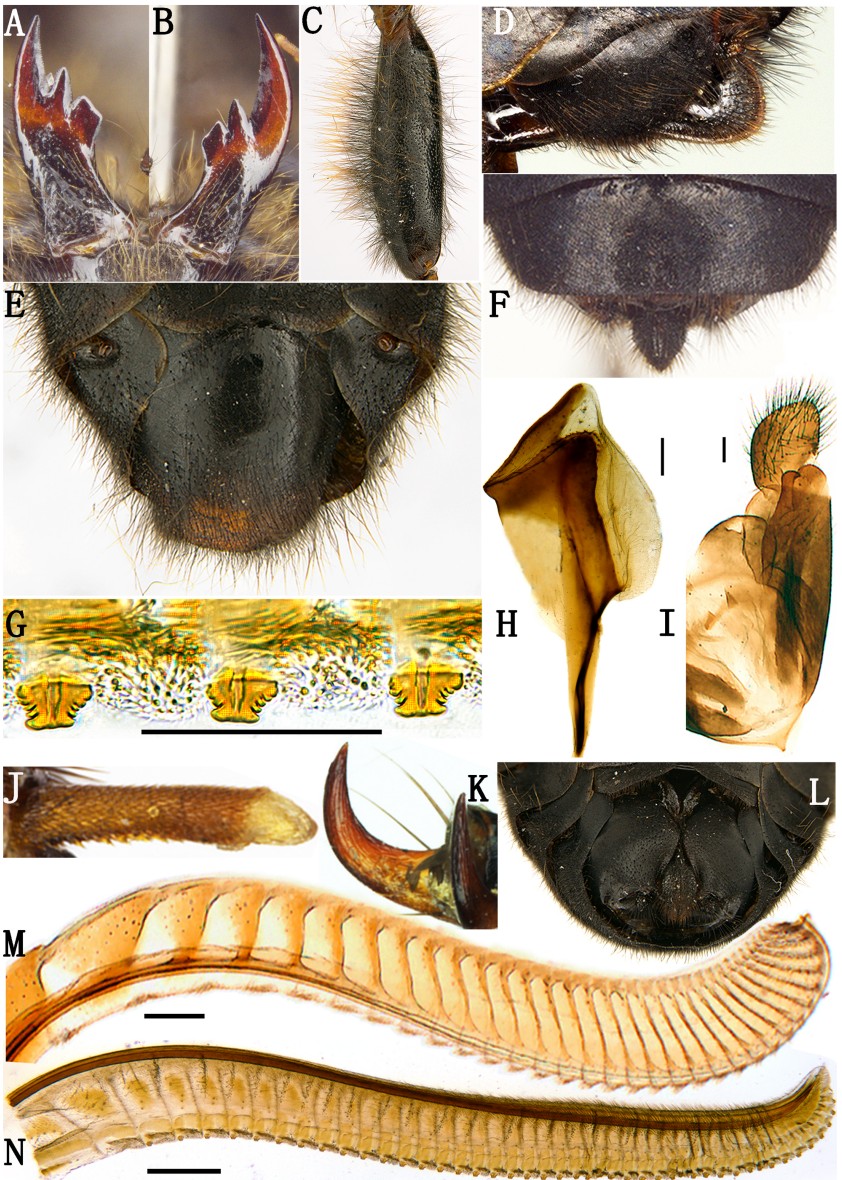

**Figure 3** *Labriocimbex sinicus* **Yan & Wei, gen. et sp. nov.** (A) Left mandible; (B) Right mandible; (C) Femur of hind leg; (D) Ovipositor sheath of female, lateral view; (E) Genital plate of male, ventral view; (F) Ovipositor sheath of female, dorsal view, (G) Middle serrulae, Scale bar = 50 um; (H) Penis valve; (I) Gonoforcep; (J) Spur of hind tibia; (K) Claw; (L) Apex of abdomen, ventral view; (M) Lance; (N) Lancet (H, I, M and N, Scale bar = 200 μm).

Key to extant Holarctic genera of Cimbicinae

1 Anal cell of fore wing divided into two by a long medial constriction; clypeus not enlarged, distinctly narrower than distance between lower corners of eyes, or antennae with 5 antennomeres and nearly as long as head breadth; inner margins of eyes strongly convergent or strongly divergent downward......................................................................................................2

– Anal cell of fore wing divided into two near the middle by a straight vein or at least touching in the middle (rarely); clypeus distinctly broader than distance between lower corners of eyes; antennae at least with 6 antennomeres (except for *Pseudoclavellaria*) and distinctly longer than head breadth; inner margin of eyes subparallel or slightly convergent downward **Cimbicinae**.................................................................................................................................. 3

2 Inner margins of eyes strongly convergent downward; distance between antennal toruli about 2 times as long as breadth of clypeus, clypeus not separated from supraclypeal area by epistomal sulcus; anterior tentorial small and shallow; head narrowed behind eyes in dorsal view, POL longer than OCL; hind orbit with distinct occipital carina; mesonotum with notauli between mesoscutal lateral lobe and middle lobe almost obsolete; hind coxae separated, apex of tibial spur acute and sclerotized. **Corynidinae**..........................................*Corynis* Thunberg

– Inner margins of eyes strongly divergent downward; distance between antennal toruli about as long as breadth of clypeus, clypeus separated from supraclypeal area by a distinct transversal furrow; anterior tentorial large and deep; head enlarged behind eyes in dorsal view, POL much shorter than OCL; hind orbit round, without occipital carina; mesonotum with notauli between mesoscutal lateral lobe and middle lobe deep; hind coxae contiguous, apex of tibial spur blunt and membranous........................................................................................ **Abiinae**

3 Labrum small, clearly narrower than 1/4 breadth of clypeus; clypeus narrower than lower distance between eyes, or clypeus triangularly convex and merging with supraclypeal area...................................................................................................................................4

– Labrum large, not narrower than 1/3 breadth of clypeus; jugum region in hind wing without crossvein; clypeus very short and much broader than lower distance between eyes, not merging with supraclypeal area................................................................................................7

4 Jugum region in hind wing without crossvein; ventral side of middle and hind femora with 1–2 rows of denticles, or anal cell in fore wing with a punctiform middle petiole, or head narrowed behind eyes in dorsal view...........................................................................5

– Jugum region in hind wing with a crossvein; middle and hind femora without denticle ventrally; anal cell of fore wing with a middle crossvein; head not narrowed behind eyes in dorsal view; Clypeus and supraclypeal area not entirely merging together and with a shallow depression between them; distance between antennal toruli and posterior margin of head about as long as distance between toruli and anterior margin of clypeus; postocellar area about as long as broad..........................................................................................*Cimbex* Olivier.

5 Outer side of middle and hind coxae with a large denticle; ventral side of hind femur with 1-2 rows of denticles; mandibles simple, without inner tooth; malar space very long, clearly longer than antennomere 4; antennae with 8–9 antennomeres [claw simple]...*Odontocimbex* Malaise

– Outer side of coxae without denticle; ventral side of femora without 1–2 rows of denticles; mandibles with distinct inner tooth; malar space short, at most as long as antennomere 1, much shorter than antennomere 4; antennae with 6–7 antennomeres........................6

6 Clypeus clearly narrower than shortest distance between eyes and separated from supra-clypeal area by a shallow but distinct furrow; anal cell in fore wing with a short petiole at basal third; claw simple; tibial spur stout, much shorter than apical breadth of tibia, apex obtuse; head enlarged behind eyes in dorsal view...............................................................***Praia*** Andre

– Clypeus clearly broader than shortest distance between eyes and merging with supraclypeal area, furrow between them absent; anal cell in fore wing with a distinct crossvein at basal fourth; claw bifurcate; tibial spur slender, clearly longer than apical breadth of tibia, tapering toward apex; head narrowed behind eyes in dorsal view.........................…................................................***Agenocimbex*** Rohwer

7 Ventral side of middle and hind femur with distinct denticle near apex [clypeus and labrum black; labrum narrowed toward base...............….......................................***Trichiosoma*** Leach

– Ventral side of femur without denticle..................................................................................8

8 Head and thorax with dense and long hairs; club of antennae not segmented; abdominal tergum 1 without lateral carina; tarsal pulvilli large, 1st and 2nd pulvilli nearly touching to each other, first pulvillus longer than 1/2 length of basitarsus; apical anal cell of forewing 1 to 2 times length of basal anal cell.…………….…………………….......………9

– Head and thorax without dense and long hairs; club of antennae distinctly segmented; abdominal tergum 1 at least with distinct lateral carina at basal 1/2; tarsal pulvilli short and small, separated each other, distance between basal 2 pulvilli not shorter than length of a pulvillus, first pulvillus much shorter than half length of basitarsus; fore wing with apical anal cell about 3 times length of basal anal cell.………...............***Leptocimbex*** Semenov

9 Labrum broadened toward base and distinctly narrowed toward apex; antennae with 6 antennomeres; inner spur of hind tibia as long as apical breadth of tibia; abdominal terga with long hairs; fore wing with apical anal cell 2 times as long as basal anal cell, vein 2m-cu almost interstitial to vein 2r-m; clypeus and labrum black..........***Labriocimbex*** Yan and Wei, gen. nov.

– Labrum clearly narrowed toward base and distinctly broadened toward apex; antennae with 5 antennomeres; inner spur of hind tibia clearly shorter than apical breadth of tibia; abdominal terga without long hairs; forewing with apical anal cell as long as basal anal cell, vein 2m-cu remote from vein 2r-m; clypeus and labrum white.........................…...***Pseudoclavellaria*** Schulz

***Labriocimbex sinicus* Yan & Wei sp. nov.** (Figs. 1–3)
urn: lsid:zoobank.org:act:E1454ED2-5321-4D39-97C2-EC8957D034C1

**Female. (Holotype)** Body length 21 mm (Fig. 1A). Black; apical 1/2 of mandible reddish brown (Figs. 3A, 3B); inner and ventral side of club of antennae largely brown, outer side dark brown (Fig. 2H); cenchri pale yellowish white; posterior half of mesepimeron, metapleuron largely, metanotum except for a small macula behind cenchri and most of metapostnotum, median triangular macula and narrow posterior margin of abdominal tergum 1 (Fig. 2E) yellowish white; apex of each tibia, tarsus and claw reddish brown,

tarsal pulvillus grayish white (Fig. 2F). Wings brownish hyaline, stigma black, basal 3/5 of vein C in fore wing, basal 3/7 of vein Sc+R and entire vein M+Cu pale yellow, vein A pale brown, other veins largely black, vein J and basal parts of all other veins in hind wing pale yellow (Fig. 1A). Hairs on face and gena black at base and yellowish white at apex (Fig. 2A); hairs on vertex of head and mesonotum black (Fig. 2D); hairs on pronotum and scutellum yellowish white largely except for black basal 0.2; hairs on mesopleuron, coxae and femora yellowish brown largely with less than basal 0.3 black (Figs. 2C, 2E, 2F); inner hairs of fore tibia reddish yellow; abdominal terga 1–2 and posterior margins of terga 3–4 with yellowish white hairs; hairs on ventral side of terga and sternites 1–3 black in basal 0.4 and reddish yellow in apical 0.6.

Body densely microsculptured, matt; lower margin of orbit, small fovea lateral to lateral ocellus, apical half of mandible, narrow lateral side of mesoscutal lateral lobe, ventral part of trochanters and of femora distinctly shiny, ventral half of mesepisternum feebly microsculptured mixed with some minute punctures, shiny; venter of abdomen feebly shiny.

Apex of labrum thickened with middle notch (Fig. 2A); median fovea round and deep, lateral foveae obscure (Fig. 2D); middle of frons concave, lateral furrow of frons shallow; postocellar furrow distinct, interocellar furrow long and deep; postocellar area quadrate, middle furrow very shallow, indistinct; lateral furrows shallow, weakly divergent backwards (Fig. 2D). Long hairs on gena clearly shorter than 1/3 head width in dorsal view. Club of antennae as long as length of antennomeres 4 and 5 combined, with obscure annular suture (Fig. 2H). Mesopleuron without middle oblique ridge (Fig. 2C); cenchrus 2.1 times broader than long, reniform (Fig. 2E). Coxae and femora with dense hairs longer than breadth of femora (Fig. 2F); inner hairs of tibia dense and short (Fig. 2F). Vein 2r in fore wing joining cell 2Rs at basal 0.4; cu-a joining cell 1M close to vein 1M (Fig. 1A). Abdominal terga 1–2 and posterior margin of terga 3–4 with dense and long hairs, other terga with sparse and short hairs. Sternites and ventral side of abdominal terga with spare and long hairs (Fig. 2F). Ovipositor sheath 0.8 times as long as metatarsomere 1 and 2 combined, apical margin roundish in lateral view (Fig. 3D), acute at apex in dorsal view (Fig. 3F). Lancet with 45 serrulae (Fig. 3N), middle serrulae as Fig. 3G, annular spine bands narrow, membranous area between serrulae roundly protruding, middle serrulae subtruncate at apex, with 5–6 proximal and 4–5 distal subbasal teeth (Fig. 3G).

**Male**: Body length 21.5 mm (Fig. 1B); body color and structure similar to female except for following parts: labrum broad and large; anterior margin of clypeus arc-shape, without incision; metathorax and abdominal tergum 1 entire black; subgenital plate slightly broader than long (Fig. 3E), apical margin round; apex of each sternite with clear middle incision, both sides roundly arcuate. Penis valve shown in Fig. 3H, gonoforcep shown in Fig. 3I.

**Holotype**. Female (CSCS13010_Lab001). China: Hunan Province, Wugang County, Mt. Yun, Yunfengge alt. 1,380 m, 26°38.630′N, 110°37.299′E, April 13, 2013, Zejian Li leg.

**Paratypes**: 17 Females, 15 Males (CSCS13010_Lab002–033). Collecting information as the holotype. 18 Females, 10 Males (CSCS13015_Lab034–061), locality and collector as the holotype, April 15, 2013. 45 Females, 17 Males (CSCS13014_Lab062–123), locality and collecting time as the holotype, Liwei Qi, Biao Chu leg. 36 Females, 51 Males

(CSCS11009_Lab124–210), China: Hunan Province, Wugang County, Mt. Yun, Shengli Temple, alt. 1,145 m, 26°38.859′N, 110°37.026′E, April 18–22, 2011, Zejian Li, Liwei Qi leg. 17 Females, 22 Males (CSCS05001_Lab211–249), China: Hunan Province, Wugang County, Mt. Yun, alt. 800–1,100 m, April 24–26, 2005, Meicai Wei, Shaobing Zhang, Wei Xiao leg. One Male (CSCS1999001_Lab250), China: Hunan Province, Wugang County, Mt. Yun, alt. 1,300 m, April 3, 1999, Wei Xiao leg. Two Females, six Males (LSAF18029_Lab251–258), China: Zhejiang Province, Lin'an City, Mt. Tianmu, alt. 1,506, 30.349° N, 119.424° E, April 19, 2018, Zejian Li, Mengmeng Liu leg. One Females (LSAF17053_Lab259), locality and collector as the former, April 16, 2017. One Females, 26 males (LSAF17054_Lab259–285), locality as the former, April 17, 2017, Tingting Ji leg. Four Females, two Males (CSCS18006_Lab286–291), China: Hunan Province, Wugang County, Mt. Yun, alt. 1,124 m, 26°38.059′N, 110°37.017′E, April 03, 2018, Meicai Wei, Gengyun Niu, Hannan Wang leg. Seven Females, one Males (CSCS18007_Lab292–299), locality as the former, alt. 1,129 m, 26°39.003′N, 110°37.027′E, April 04, 2018, Meicai Wei, Hannan Wang leg.

**Variation.** Body length 18–24 mm in female, 19–24 mm in male; club of antennae color brown to pale yellowish brown; hairs color on pronotum and scutellum yellowish white to yellowish brown.

**Distribution**. China (Hunan, Zhejiang).

**Etymology:** The specific name of the new species refers to the distribution area, China.

**Remarks.**

*Labriocimbex pilosus* sp. nov. (*Li & Wu, 2010*) and *Labriocimbex sinicus* (*Vilhelmsen, 2019*) were two nomina nuda and have never been properly described before this paper. The two manuscript names were originally proposed by the senior author of this paper (MW) for the two undescribed species found in China. The former species represents only by a few specimens from different localities and so it is not described here, and specimen collection record: one female, China, Gansu Province, Mt. Xiaolong, Maiji forest farm, Sun hill; alt. 1,620, 34°25′11.0″N, 105°46′30.1″E, April 17, 2009, Wu XingYu leg.

*Labriocimbex sinicus* Yan & Wei, sp. nov. is similar to *L. zaraeoides* (*Malaise, 1939*) comb. nov. (Fig. 4A), but differs from the latter in the following characters: the clypeal notch deep, depth about 1/2 length of clypeus; between the clypeus and supraclypeal area with a distinct transverse furrow; the long hairs on gena 3.5 times as long as diameter of lateral ocellus, longer than the shortest axis of an eye; the long hairs on mesopleuron about 4.5 times as long as diameter of lateral ocellus; the abdominal tergum 1 largely black.

**Labriocimbex zaraeoides** (*Malaise, 1939*) **comb. nov**. (Fig. 4)
*Trichiosoma zaraeoides Malaise, 1939*: 16–17.

**Distribution**. Northern Myanmar.

**Remarks.** This species is similar to *L. sinicus* Yan & Wei sp. nov., the majority of the characters place it in the new genus *Labriocimbex.* The most important of these characters are: the broadly emarginated clypeus, the triangular labrum (Fig. 4C), the form of the antennae (antennae with 5 antennomeres before the rigid club; joints of the club very

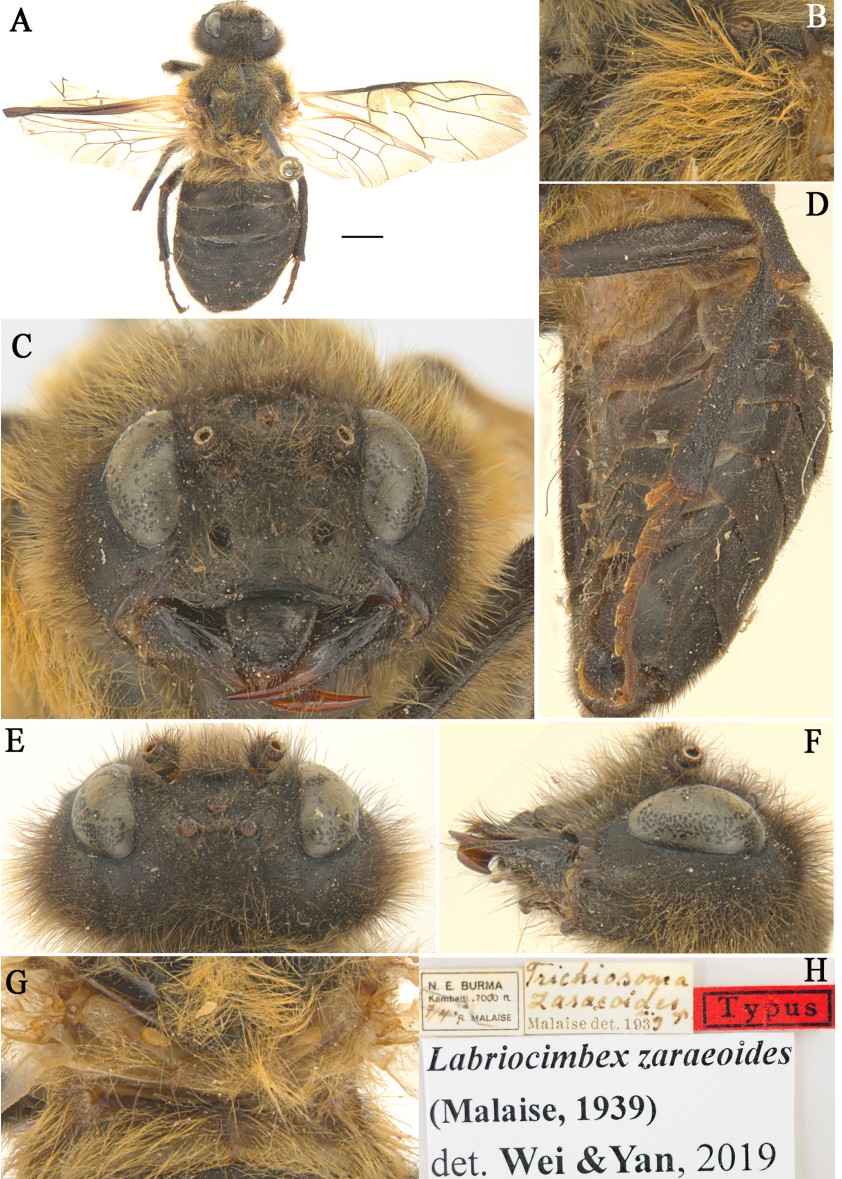

**Figure 4** *Labriocimbex zaraeoides Malaise, 1939*) **comb. nov.** (A) Adult female (holotype), Scale bar = 2 mm; (B) Mesopleuron of female, lateral view; (C) Head of female, front view; (D) Abdomen, lateral view; (E) Head of female, dorsal view; (F) Head of female, lateral view; (G) Metanotum and basal of abdominal terga, dorsal view; (H) Labels.

indistinct [sic!] *Malaise, 1939*); the slender hind legs (the coxae and femur of leg with long hairs; the ventral side of hind femur without a denticle near apex; the base of hind tibia narrower than apex; the 1st and 2nd tarsal pulvilli long, Fig. 4D) the venation (the anal crossvein punctiform and the anal cell strongly narrowed, Fig. 4A); the color of body (the posterior half of mesepimeron, metapleuron, metanotum and abdominal tergum 1 yellowish white, Figs. 4B, 4D and 4G). The characters that separate this species from all known *Trichiosoma* are the yellowish white color of the metanotum and the base of
the abdomen, and the ventral side of the hind femur without a large denticle near apex. *L. zaraeoides* differs from *L. sinicus* in the following characters: the clypeal notch shallow, depth about 1/4 length of clypeus; the transverse furrow between clypeus and supraclypeal area absent (Fig. 4C); the long hairs on gena 2.5 times the diameter of lateral ocellus, shorter than the shortest axis of an eye (Fig. 4E); the long hairs on mesopleuron about 3.5 times as long as diameter of the lateral ocellus (Fig. 4B); the abdominal tergum1 largely yellow brown (Fig. 4G).

## General features of the *L. sinicus* mitochondrial genome

We sequenced the complete mitochondrial genome of *L. sinicus* (GenBank accession no. MH136623), a typical set of 37 genes, including 13 PCGs, 22 tRNAs and two rRNAs. Most of the genes were located on the J- strand except for four PCGs, two rRNAs and seven tRNAs (Table 4).

A total of 14 pairs of genes were directly adjacent, without overlapping or intergenic nucleotides. The total length of the intergenic regions was 268 bp in 18 locations with a size ranging from 1 to 50 bp (Table 4). The longest was located between *trnH* and *ND4*, while the second longest was 45 bp located between *rrnS* and *trnM*. In comparison with the mitochondrial genome of *T. anthracinum* and *C. lateralis*, there were differences in the length of intergenic spacers and locations. The longest (414 bp) was located at the start of the mitochondrial genome before *trnY* in *T. anthracinum*. The longest length of the intergenic spacers was 345 bp located between the *ND6* and *CYTB* genes in *C. lateralis*. We found that homologous searches on the longest intergenic region of *L. sinicus* revealed substantial differences from any identified Symphyta sequence.

There were in total 32 overlapping nucleotides between neighboring genes in six locations, and the range of length of the overlapping sequence is from 3 to 14 bp: *trnM* and *trnQ*, *ATP8* and *ATP6*, *ND4* and *ND4L*, *trnN* and *trnS2*, and *ATP6* and *COIII*; and the longest was 14 bp between *ATP6* and *COIII* (Table 4). The common motifs such as: ATGATAA between *ATP8* and *ATP6*, and ATGTTAA between *ND4* and *ND4L*, which also exist in *T. anthracinum*, and are not found in *C. lateralis*, are common features of many other insect mitochondrial genomes (*Song et al., 2016*; *Doğan & Korkmaz, 2017*).

## Protein-coding genes and codon usage

The mitochondrial genome of *L. sinicus* contains 13 PCGs, and its length is 12,456 bp, accounting for 80.86% of the total length (Table 5). All PCGs were initiated by ATN codons. All PCGs were ended with TAA as stop codon except for *ND5* which ended with T (Table 4).

The codon usage of *L. sinicus* also shows a significant bias towards A/T Leu, Ile, Phe and Ser, were found as the most frequently used amino acids. TTA-Leu showed the highest RSCU of 5.04 (Table 6). Comparisons of the RSCU with those of *C. lateralis* and *T. anthracinum* showed a similar pattern for codon usage bias and reflected a significant correlation between codon preference and nucleotide composition, that is similar to other symphytan species (*Dowton et al., 2009a*; *Wei et al., 2010*; *Wei, Wu & Liu, 2015*; *Korkmaz et al., 2015*; *Korkmaz et al., 2016*; *Korkmaz et al., 2017*; *Song et al., 2015*; *Song et al., 2016*; *Niu*

**Table 4  Mitochondrial genome characteristics of _L. sinicus._.**

| Gene | Strand | Start | Stop | Length(bp) | Start codon | Stop codon | Anticodon | IGN |
|------|--------|-------|------|-----------|-------------|------------|-----------|-----|
| _trnI_ | J | 1 | 67 | 67 | | | GAU | 1 |
| _ND2_ | J | 70 | 1113 | 1,044 | ATG | TAA | | 2 |
| _trnW_ | J | 1117 | 1181 | 65 | | | UCA | 3 |
| _COI_ | J | 1182 | 2720 | 1,539 | ATT | TAA | | 0 |
| _trnL2_ | J | 2760 | 2825 | 66 | | | UAA | 39 |
| _COII_ | J | 2827 | 3510 | 684 | ATG | TAA | | 1 |
| _trnK_ | J | 3532 | 3602 | 71 | | | CUU | 21 |
| _trnD_ | J | 3603 | 3672 | 70 | | | GUC | 0 |
| _ATP8_ | J | 3673 | 3834 | 162 | ATC | TAA | | 0 |
| _ATP6_ | J | 3828 | 4517 | 690 | ATG | TAA | | −7 |
| _COIII_ | J | 4504 | 5289 | 786 | ATG | TAA | | −14 |
| _trnG_ | J | 5310 | 5373 | 64 | | | UCC | 20 |
| _ND3_ | J | 5374 | 5724 | 351 | ATT | TAA | | 0 |
| _trnA_ | J | 5732 | 5797 | 66 | | T | UGC | 7 |
| _trnR_ | J | 5798 | 5,864 | 67 | | | UCG | 0 |
| _trnN_ | J | 5,866 | 5,934 | 69 | | | GUU | 1 |
| _trnS1_ | J | 5,935 | 6,002 | 68 | | | UGA | 0 |
| _trnE_ | J | 6,010 | 6,076 | 67 | | | UUC | 7 |
| _trnF_ | N | 6,092 | 6158 | 67 | | | AAG | 15 |
| _ND5_ | N | 6,159 | 7872 | 1,714 | ATT | T | | 0 |
| _trnH_ | N | 7873 | 7940 | 68 | | | GUG | 0 |
| _ND4_ | N | 7991 | 9343 | 1,353 | ATT | TAA | | 50 |
| _ND4L_ | N | 9337 | 9618 | 282 | ATT | TAA | | −7 |
| _trnT_ | N | 9621 | 9865 | 65 | | | UGU | 2 |
| _trnP_ | N | 9686 | 9751 | 66 | | | GGU | 0 |
| _ND6_ | J | 9753 | 10256 | 504 | ATA | TAA | | 1 |
| _CYTB_ | J | 10258 | 11391 | 1,134 | ATA | TAA | | 1 |
| _trnS2_ | J | 11435 | 11502 | 68 | | | UCU | 43 |
| _ND1_ | N | 11512 | 12462 | 951 | ATT | TAA | | 9 |
| _trnL1_ | N | 12463 | 12530 | 68 | | | GAU | 0 |
| _rrnL_ | N | 12531 | 13871 | 1,341 | | | | 0 |
| _trnV_ | N | 13872 | 13941 | 70 | | | CAU | 0 |
| _rrnS_ | N | 13941 | 14731 | 791 | | | | −1 |
| _trnM_ | J | 14777 | 14,845 | 69 | | | CAU | 45 |
| _trnQ_ | N | 14843 | 14,911 | 69 | | | GUU | −3 |
| AT-rich region | none | 14912 | 15,261 | 350 | | | | 0 |
| _trnY_ | J | 15262 | 15331 | 70 | | | GUA | 0 |
| _trnC_ | N | 15333 | 15403 | 71 | | | ACG | 1 |

_et al., 2019a_; _Niu et al., 2019b_; _Du et al., 2018_; _Ma et al., 2019_; _Tang et al., 2019_). Codons rich in C and G, CGC-Arg and CTC-Leu was absent, CGG-Arg, GGC-Gly, AGC-Ser, ACG-Thr, CTG-Leu, GTC-Val, GTG-Val and TGC-Cys, were used once, AGG-Ser, TCG-Ser, TCC-Ser, CCG-Pro and GCG-Ala were rarely used, which is similar to both

**Table 5** Nucleotide composition of *L. sinicus* mitochondrial genome.

| Feature | Length(bp) | A% | C% | G% | T% | A+T% | AT-skew | GC-skew |
|---|---|---|---|---|---|---|---|---|
| Whole genome | 15,405 | 43.5 | 11.1 | 7.7 | 37.7 | 81.2 | 0.0714 | −0.1809 |
| Protein coding genes | 12,456 | 34.4 | 9.7 | 10.4 | 45.5 | 79.9 | −0.1389 | 0.0348 |
| First codon position | 4,152 | 36.9 | 9.5 | 15.1 | 38.5 | 75.4 | −0.0212 | 0.2276 |
| Second codon position | 4,152 | 20.9 | 16.2 | 13 | 49.9 | 70.8 | −0.4096 | −0.1096 |
| Third codon position | 4,152 | 45.5 | 3.5 | 3 | 48 | 93.5 | −0.0267 | −0.0769 |
| Protein coding genes-J | 6,840 | 37.8 | 12 | 9.3 | 40.9 | 78.7 | −0.0394 | −0.1268 |
| First codon position | 2,280 | 40.4 | 11.9 | 14.6 | 33.1 | 73.5 | 0.0993 | 0.1019 |
| Second codon position | 2,280 | 23 | 18.7 | 12 | 46.3 | 69.3 | −0.3362 | −0.2182 |
| Third codon position | 2,280 | 50 | 5.3 | 1.4 | 43.3 | 93.3 | 0.0718 | −0.5821 |
| Protein coding genes-N | 5,616 | 30.3 | 7 | 11.6 | 51.1 | 81.4 | −0.2555 | 0.2473 |
| First codon position | 1,872 | 32.5 | 6.7 | 15.6 | 45.2 | 77.7 | −0.1634 | 0.3991 |
| Second codon position | 1,872 | 18.4 | 13.1 | 14.3 | 54.3 | 72.7 | −0.4938 | 0.0438 |
| Third codon position | 1,872 | 40.1 | 1.2 | 5 | 53.7 | 93.8 | −0.1450 | 0.6129 |
| ATP6 | 690 | 38.3 | 11.2 | 8 | 42.6 | 80.9 | −0.0532 | −0.1667 |
| ATP8 | 162 | 45.1 | 9.3 | 2.5 | 43.2 | 88.3 | 0.0215 | −0.5763 |
| ND1 | 951 | 51.4 | 12.3 | 6.9 | 29.3 | 80.7 | 0.2739 | −0.2813 |
| ND2 | 1044 | 44.1 | 9.9 | 5.7 | 40.3 | 84.4 | 0.0450 | −0.2692 |
| ND3 | 351 | 35 | 10.5 | 9.7 | 44.7 | 79.7 | −0.1217 | −0.0396 |
| ND4 | 1353 | 51.2 | 11.5 | 7.4 | 29.9 | 81.1 | 0.2626 | −0.2169 |
| ND4-BLASTP | 1344 | 51.3 | 11.6 | 7.4 | 29.7 | 81 | 0.2667 | −0.2211 |
| ND4L | 282 | 49.6 | 12.1 | 3.5 | 34.8 | 84.4 | 0.1754 | −0.5513 |
| ND5 | 1714 | 50.8 | 11.1 | 6.8 | 31.3 | 82.1 | 0.2375 | −0.2402 |
| ND6 | 504 | 42.1 | 8.7 | 5 | 44.2 | 86.3 | −0.0243 | −0.2701 |
| COI | 1539 | 35.1 | 13.5 | 12.8 | 38.7 | 73.8 | −0.0488 | −0.0266 |
| COII | 684 | 40.8 | 12.7 | 8 | 38.5 | 79.3 | 0.0290 | −0.2271 |
| COIII | 786 | 33.5 | 13 | 12 | 41.6 | 75.1 | −0.1079 | −0.0400 |
| CYTB | 1134 | 35.4 | 13.1 | 10.4 | 41.2 | 76.6 | −0.0757 | −0.1149 |
| 12s | 791 | 44 | 10.7 | 5.3 | 40.1 | 84.1 | 0.04637337 | 0.3375 |
| 16s | 1341 | 46.8 | 11 | 4.9 | 37.4 | 84.2 | 0.111639 | −0.383648 |

cimbicid mitochondrial genomes (Table 6). The ratio can be calculated by rate of G + C rich codons (Pro, Ala, Arg, and Gly) and A + T rich codons (Phe, Ile, Met, Tyr, Asn, and Lys), and it is 0.28 in *L. sinicus*, which is similar to those of other symphytan species (0.28–0.31) (*Korkmaz et al., 2015*). The translation, initiation, and termination signals as well as the codon usage of the *L. sinicus* mitochondrial genome do not display any unusual characteristics (Table 6).

## Gene rearrangement and nucleotide composition

The mitochondrial genome of symphytan species appears to be more conserved than that of Apocrita (*Song et al., 2016*; *Wei, Niu & Du, 2014*). However, compared with the putative ancestral mitochondrial genome of insects, we detected several rearrangement events in three tRNA gene clusters in *L. sinicus* (Fig. 5), The first rearrangement event is found in the clusters of *trnI-trnQ-trnM*, where *trnM* and *trnQ* was founding swapped positions, in

**Table 6  Codon usage of PCGs in mitochondrial genome of *L. sinicus*. No., frequency of each codon; RSCU, relative synonymous condon usage.**

| Amino acid | Codon | NO. | RSCU | Amino acid | Codon | NO. | RSCU |
|---|---|---|---|---|---|---|---|
| Phe | TTT | 409 | 1.9 | Tyr | TAT | 159 | 1.78 |
|  | TTC | 21 | 0.1 |  | TAC | 20 | 0.22 |
| Leu | TTA | 560 | 5.04 | End | TAA | 0 | 0 |
|  | TTG | 35 | 0.31 |  | TAG | 0 | 0 |
| Leu | CTT | 37 | 0.33 | His | CAT | 68 | 1.79 |
|  | CTC | 0 | 0 |  | CAC | 8 | 0.21 |
|  | CTA | 34 | 0.31 | Gln | CAA | 61 | 1.85 |
|  | CTG | 1 | 0.01 |  | CAG | 5 | 0.15 |
| Ile | ATT | 464 | 1.87 | Asn | AAT | 237 | 1.84 |
|  | ATC | 31 | 0.13 |  | AAC | 20 | 0.16 |
| Met | ATA | 314 | 1.91 | Lys | AAA | 135 | 1.88 |
|  | ATG | 15 | 0.09 |  | AAG | 9 | 0.13 |
| Val | GTT | 83 | 2.21 | Asp | GAT | 62 | 1.82 |
|  | GTC | 1 | 0.03 |  | GAC | 6 | 0.18 |
|  | GTA | 65 | 1.73 | Glu | GAA | 72 | 1.85 |
|  | GTG | 1 | 0.03 |  | GAG | 6 | 0.15 |
| Ser | TCT | 134 | 2.67 | Cys | TGT | 37 | 1.95 |
|  | TCC | 4 | 0.08 |  | TGC | 1 | 0.05 |
|  | TCA | 116 | 2.31 | Trp | TGA | 92 | 1.8 |
|  | TCG | 2 | 0.04 |  | TGG | 10 | 0.2 |
| Pro | CCT | 64 | 1.97 | Arg | CGT | 20 | 1.54 |
|  | CCC | 15 | 0.46 |  | CGC | 0 | 0 |
|  | CCA | 48 | 1.48 |  | CGA | 31 | 2.38 |
|  | CCG | 3 | 0.09 |  | CGG | 1 | 0.08 |
| Thr | ACT | 70 | 1.74 | Ser | AGT | 23 | 0.46 |
|  | ACC | 8 | 0.2 |  | AGC | 1 | 0.02 |
|  | ACA | 82 | 2.04 |  | AGA | 119 | 2.37 |
|  | ACG | 1 | 0.02 |  | AGG | 2 | 0.04 |
| Ala | GCT | 65 | 2.08 | Gly | GGT | 62 | 1.22 |
|  | GCC | 7 | 0.22 |  | GGC | 1 | 0.02 |
|  | GCA | 49 | 1.57 |  | GGA | 112 | 2.2 |
|  | GCG | 4 | 0.13 |  | GGG | 29 | 0.57 |

addition, *trnM-trnQ* was translocated from the *trnI-trnQ-trnM* cluster to a downstream position of *rrnS*; which have not been reported for any symphytan mitogenome to date. The second event is corresponding to the remote inversion of *trnY* and the translocation of *trnC* from a location between *trnW* and *COX1* to upstream of *trnI*, which has great similarity to the gene order and rearrangement events observed in *T. anthracinum*. The arrangement of cluster of *trnW-trnC-trnY* appears to be mostly conserved in almost all known symphytan mitogenomes, except for representative cimbicid species. The last event is only found in the TP cluster of *L. sinicus*, and here *trnT* is inverted. The gene order from *COI* to *rrnS* is conserved in all sequenced species of Cimbicidae.

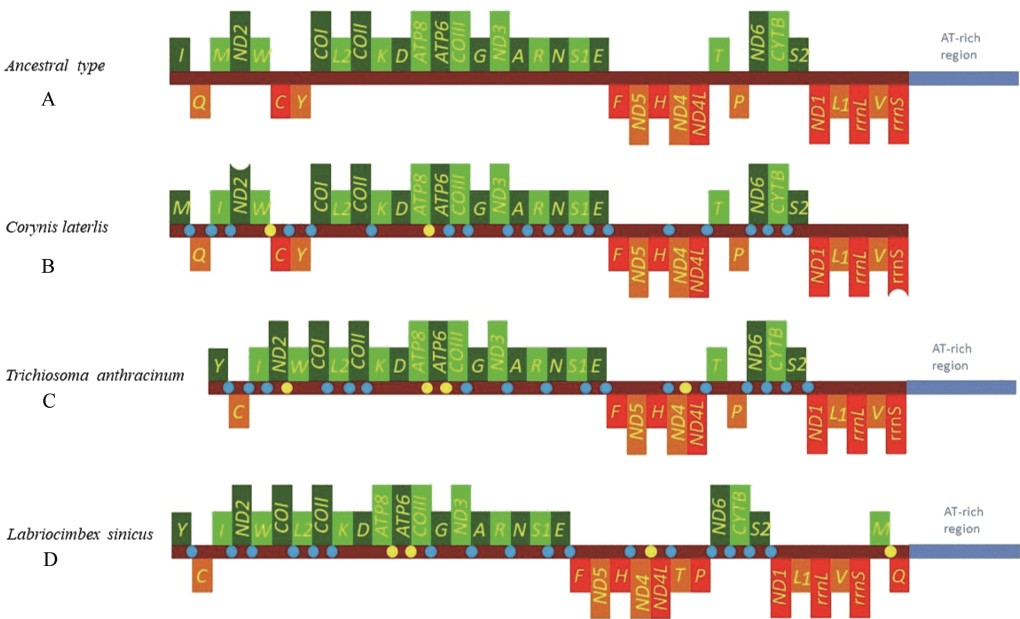

**Figure 5** **Mitochondrial genome organization of three cimbicid species referenced with the ancestral insect mitochondrial genomes.** Genes transcribed from the J and N strands are shown with green and orange color, respectively. Overlapping and intergenic regions are marked in yellow and blue circles. tRNA genes are denoted by a one-letter symbol according to the IPUC-IUB single-letter amino acid codes A + T-rich region is marked in blue and tRNA genes are labelled by the single-letter amino acid code. (A) Ancestral type of insect mitochondrial genomes; (B) *Corynis laterlis* mitochondrial genomes; (C) *Trichiosoma anthracinum* mitochondrial genomes; (D) *Labriocimbex sinicus* mitochondrial genomes.

Similar to previously reported symphytan mitochondrial genomes (*Ma et al., 2019*; *Doğan & Korkmaz, 2017*; *Song et al., 2016*), the nucleotide compositions of *L. sinicus* (43.5% A, 37.7% T, 7.7% G and 11.1% C) were biased towards A and T, with an average 81.2% A+T content; a stronger AT bias was found in the N strand (81.4% A+T content) than in the J- strand (78.7%) (Table 5).

Further analysis of the PCGs indicated that the third codon position demonstrates the highest A + T content (93.5%), in agreement with symphytan mitochondrial genomes (*Ma et al., 2019*; *Doğan & Korkmaz, 2017*; *Song et al., 2016*). The gene with the highest A + T content was *ATP8* with 88.3% (Table 5). Here we observed that the AT-skew was slightly positive (0.0714), and the GC-skew was negative (−0.1809) when considering the whole genome (Table 5). This indicates that the occurrence of A is higher than that of T, and the occurrence of C is higher than that of G, which is a general phenomenon observed in all reported symphytan mitochondrial genomes, except for those of *Tremex columba* and *Xiphydria* sp. (*Ma et al., 2019*; *Doğan & Korkmaz, 2017*; *Song et al., 2016*; *Wei, Wu & Liu, 2015*; *Castro & Dowton, 2005*; *Dowton et al., 2009a*). However, a deviation was found in the PCGs of *L. sinicus*, in terms of AT-skew (−0.1389) and GC-skew (0.0348), which also occurred in both *C. lateralis* and *T. anthracinum*. This deviation can exert influences on the selection forces acting on the PCG codon positions, in accordance with study by *Korkmaz et al. (2015)*.

## Transfer RNA genes

In the mitochondrial genome of *L. sinicus* 15 tRNAs were encoded by the J- strand, while the remaining tRNAs were encoded by the opposite N-strand. All tRNAs folded into a common clover-leaf structure, except *trns1*-AGN, where the dihydrouridine (DHU) arm was missing (Fig. 6). The size of the tRNAs ranged from 64 bp (*trnG*) to 71 bp (*trnC, trnK*), and this usually depends on the length of the variable loop, T ΨC loop and D-loops (*Clary & Wolstenholme, 1985*). The DHU arm was 3–4 bp, the AC arm was 4–5 bp, and the T ΨC arm varied from 4–5 bp, while the amino acid acceptor (AA) stem and anticodon (AC) loops were conserved at 7 bp in all of the tRNA genes.

In the mitochondrial tRNA secondary structures, mismatches mainly occur in the DHU arm, AA arm and AC arm, and sometimes in the T ΨC arm. A total of 16 unmatched base pairs were scattered among the following tRNA genes, including 12 G-U mismatched pairs occurring in *trnA, trnD, trnQ, trnG, trnH, trnL1, trnP, trnF*, and *trnY*, and four U-U mismatches occurring in *trnR, trnT* and *trnL1*. The number of mismatches were 24 (12 G–U pairs, five U–U pairs, three A–A pairs, two A–C pairs, one A–G pair and one C–U pair) in *C. lateralis* (Fig. 7), and 18 (15 G–U pairs, two U–U pairs and 1 A–C pair) in *T. anthracinum* (Fig. 6), which is typical for Hymenoptera (*Ma et al., 2019*; *Castro & Dowton, 2005*; *Dowton et al., 2009b*). The phenomenon of aberrant mismatches, loops, or extremely short arms for tRNAs has been shown to be common in metazoan mitochondrial genomes (*Wolstenholme, 1992*).

In addition, there were some tRNA structural differences between *L. sinicus* and *T. anthracinum* (Fig. 6). The identified anticodons were almost identical to those of the cimbicid species, with the exception of the anticodon of *trnS1* (*AGn*), which is UCU in *L. sinicus* and *T. anthracinum*, as well as this is true of all previously reported of Symphyta (*Ma et al., 2019*; *Doğan & Korkmaz, 2017*; *Song et al., 2016*; *Castro & Dowton, 2005*; *Dowton et al., 2009b*).

## Ribosomal RNA genes

The *rrnL* gene of *L. sinicus* was 1,341 bp in length with an 84.2% A+T content, while *rrnS* was 791 bp in length with an 84.1% A+T content (Table 5). This was in a comparable range to homologous genes in *T. anthracinum* (1,351 bp; 800 bp) and *C. lateralis* (1,359 bp; 493 bp *rrnS* partial gene), and also identical to all reported hymenopteran species (*Gillespie et al., 2006*; *Wei et al., 2010*; *Doğan & Korkmaz, 2017*; *Song et al., 2016*; *Korkmaz et al., 2015*). Both genes were encoded on the N-strand (Table 4).

Similar to the known symphytan mitochondrial genomes, the *rrnL* gene is positioned between *trnL1* and *trnV* in three species of Cimbicidae (Fig. 5). The predicted structure of *rrnL* in *L. sinicus* is consistent with the observed pattern in *C. lateralis* and *T. anthracinum*, whereby 45 helices belonging to five domains were identified in those species (Figs. 8, 9). Domain III is absent as in other arthropods (*Korkmaz et al., 2015*), and domain II is variable in base composition, forming a long stem with a big loop structure in the area II terminal. Domains IV and V are more conserved within the Tenthredinidae than domains I, II and VI. Eight helices (H563, H579, H777, H822, H2023, H2043, H2455 and H2547) of *rrnL* are highly conserved. The H183, H991, H1057, H1196 and H2077 helices display

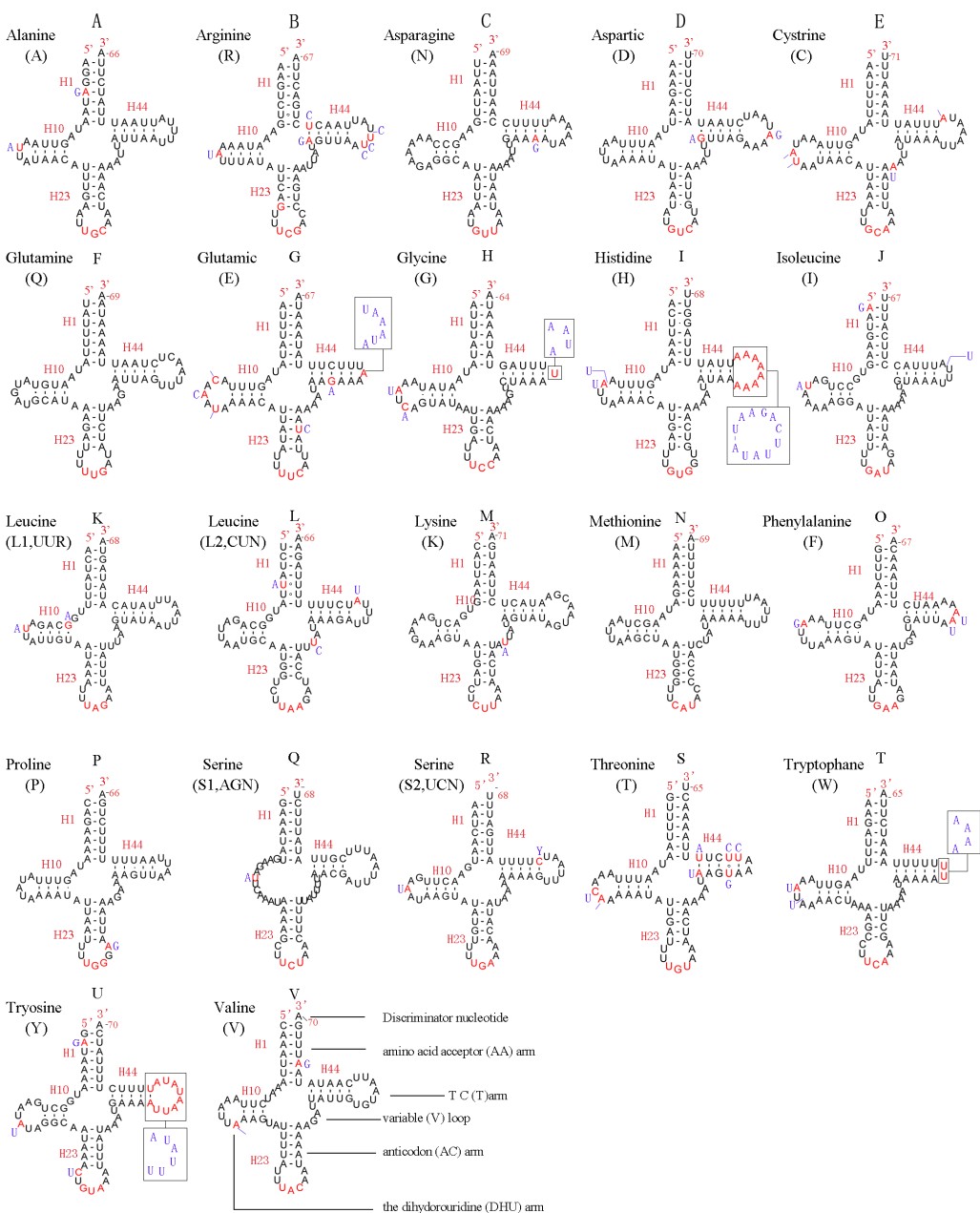

**Figure 6** **Predicted secondary structures for the 22 typical tRNA genes of *L. sinicus* and *T. anthracinum*** (adapted from *Doğan & Korkmaz, 2017*) **mitogenomes.** Base-pairing is indicated as follows: Watson–Crick pairs by lines, wobble GU pairs by dots and other noncanonical pairs by circles. Variable regions are presented in boxes with red (*L. sinicus*) and blue (*T. anthracinum*) color . (A) *trnA*; (B) *trnR*; (C) *trnN*; (D) *trnD*; (E) *trnC*; (F) *trnQ*; (G) *trnE*; (H) *trnG*; (I) *trnH*; (J) *trnI*; (K) *trnL1*; (L) *trnL2*; (M) *trnK*; (N) *trnM*; (O) *trnF*; (P) *trnP*; (Q) *trnS1*; (R) *trnS2*; (S) *trnT*; (T) *trnW*; (U) *trnY*; (V) *trnV*.

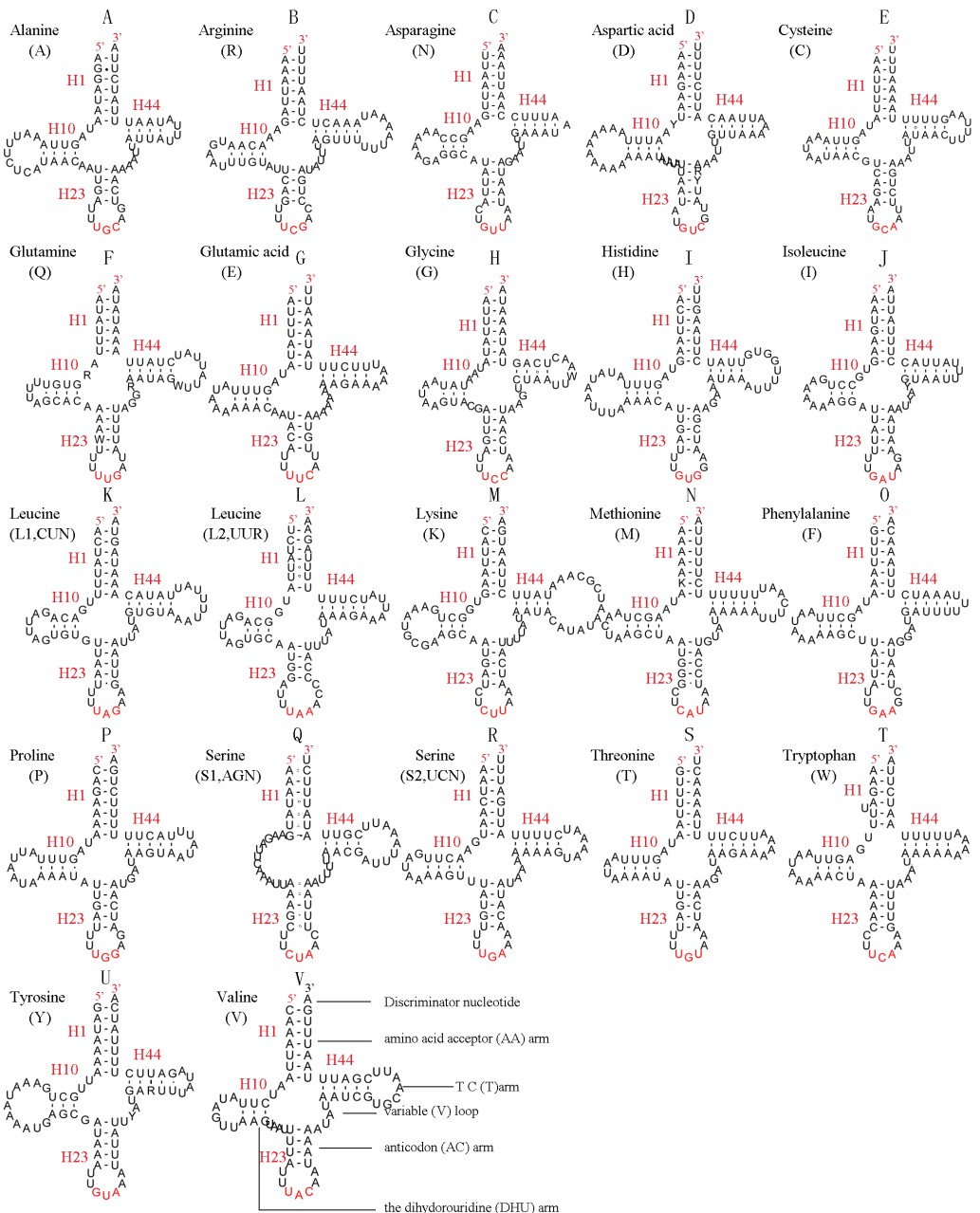

**Figure 7** **Predicted secondary structures for the 22 tRNA genes of C. *lateralis* (adapted from *Song et al., 2016*)).** Dashes indicate Watson-Crick base pairing and dots indicate G-U base pairing. (A) *trnA*; (B) *trnR*; (C) *trnN*; (D) *trnD*; (E) *trnC*; (F) *trnQ*; (G) *trnE*; (H) *trnG*; (I) *trnH*; (J) *trnI*; (K) *trnL1*; (L) *trnL2*; (M) *trnK*; (N) *trnM*; (O) *trnF*; (P) *trnP*; (Q) *trnS1*; (R) *trnS2*; (S) *trnT*; (T) *trnW*; (U) *trnY*; (V) *trnV*.

helical length and loop size/structure variability within three cimbicid *rrnL* genes (Figs. 8, 9).

The *rrnS* secondary structure of *L. sinicus* is between *trnV* and an AT-rich region, and contains four domains and 26 helices. Compared with *T. anthracinum,* it is significantly

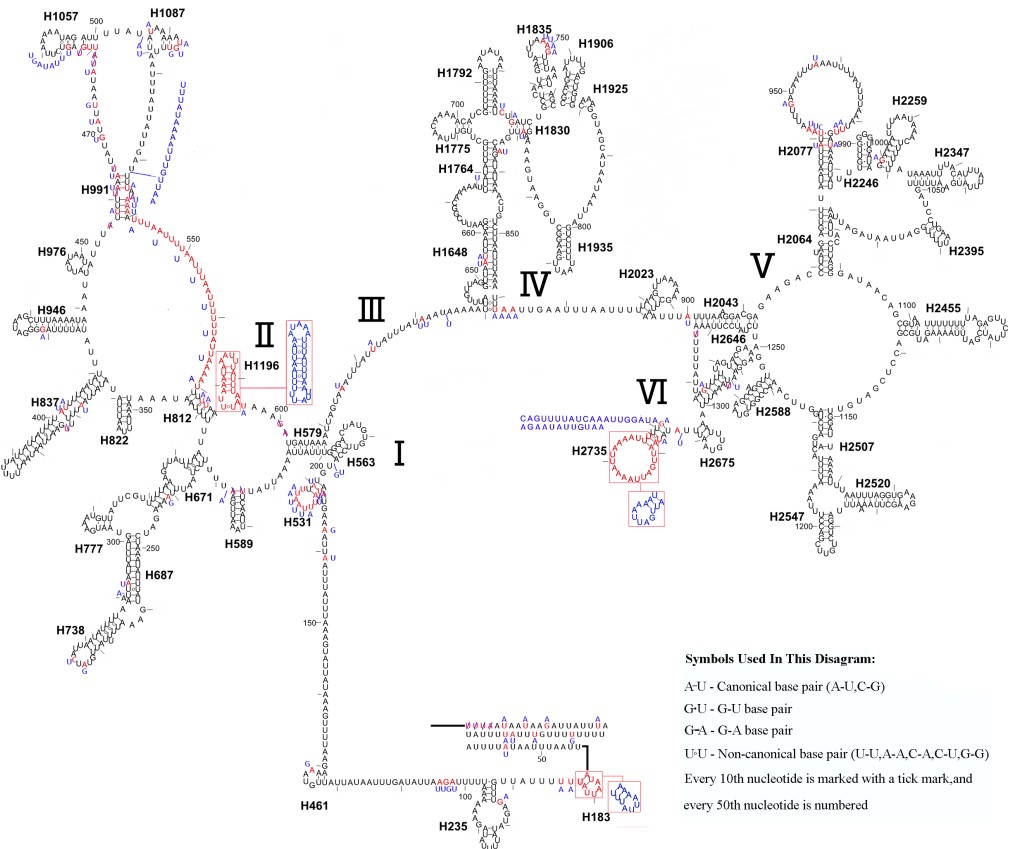

**Figure 8** **The predicted secondary structures of *rrnL* of *L. sinicus* and *T. anthracinum*.** Tertiary interactions and base triples are connected by continuous lines. The numbering of helix follows *Gillespie et al. (2006)*. Roman numbers refer to domain names. Dashes indicate Watson-Crick base pairing and dots indicate G-U base pairing. The helical variation among cimbicid species are presented in boxes with red (*L. sinicus*) and blue (*T. anthracinum*) color.

different in terms of base composition in domain II (Fig. 10). Specifically, H47 is variable among the different hymenopteran species, having a large loop. The loop size is variable and determined by overall *rrnS* length, except for in the cephid species (*Gillespie et al., 2006*; *Wei et al., 2010*; *Doğan & Korkmaz, 2017*; *Song et al., 2016*; *Korkmaz et al., 2015*). The structures of domains I and II of *C. lateralis* are missing, so they cannot be compared with those of *L. sinicus*, but the structures are similar in domains III and IV (Fig. 11). In *rrnS*, domain III and domain VI were more conserved within Tenthredinidae than domains I and II (Figs. 10 and 11).

## Phylogenetic relationships

Phylogenetic relationships within the "Symphyta" were reconstructed using both BI and ML analyses. The topologies of the two phylogenetic trees were almost identical, thus we combined the two phylogenetic trees (Fig. 12). The clade consisting of (Tenthredinidae + Cimbicidae) + (Argidae + Pergidae), was very stable with the highest nodal supports.

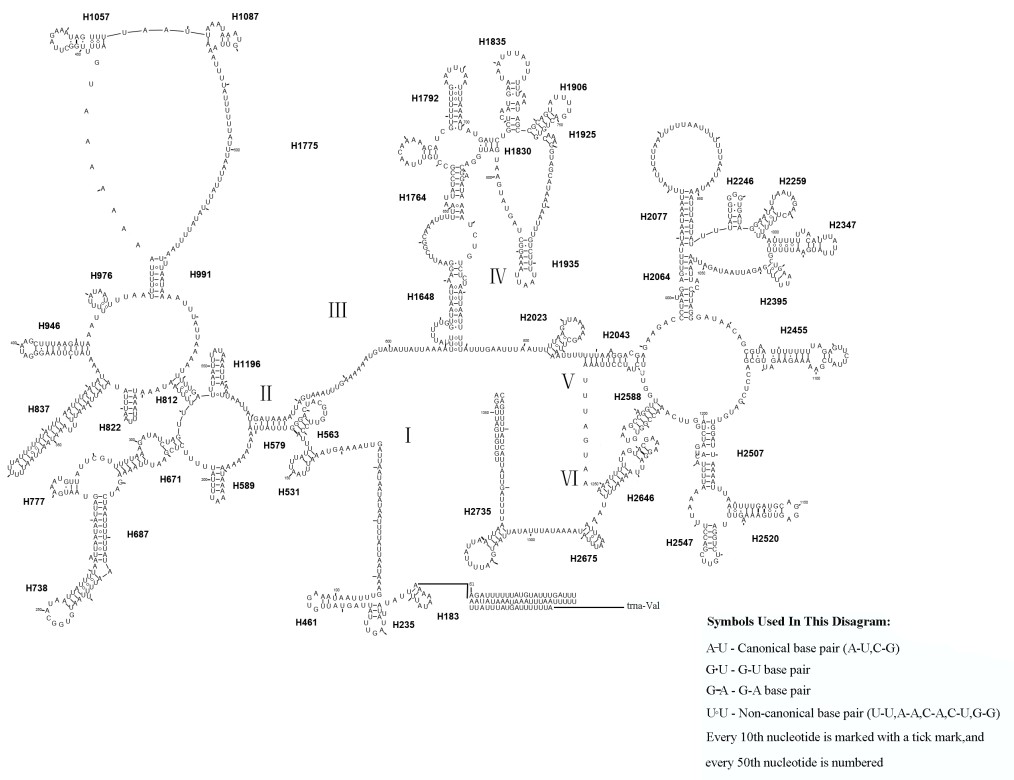

**Symbols Used In This Disagram:**

A·U - Canonical base pair (A-U,C-G)

G·U - G-U base pair

G·A - G-A base pair

U·U - Non-canonical base pair (U-U,A-A,C-A,C-U,G-G)

Every 10th nucleotide is marked with a tick mark,and

every 50th nucleotide is numbered

**Figure 9   *Corynis lateralis rrnL.*** Predicted *rrnL* secondary structure in *C. lateralis.* The numbering of helix follows *Gillespie et al. (2006)*.Roman numbers refer to domain names.

The recovered trees supported a relationship consisting of Xyelidae + (Tenthredinoidea + (Pamphiliidae + ((Megalodontesidae + Xiphydriidae) + (Cephidae + (Orussidae + (Siricidae + Apocrita)))))) in Hymenoptera. It seems unusual that Siricidae is the sister group to Apocrita, although, the sister relation of *Tremex columba* (MH422968) and *Tremex columba* (AY206795) according to the phylogenetic inference based on IV and V areas of *rrnL* (YY Zhang, 2019, unpublished data), suggests the former is valid. It showed that more mitochondrial genomes should be involved to evaluate the correctness of these unusual sister groups.

To investigate the phylogenetic relationships of *Labriocimbex* within Cimbicidae, we analyzed 43 cimbicid and two outgroups sequences, an approximately 850-bp piece of the *COI* gene were obtained in this study, combined with the other species sequences from GenBank aligned by MAFFT. The phylogenetic trees were reconstructed using both BI and ML analyses (Fig. 13). We found that the generic relationships of Cimbicidae revealed by the phylogenetic analyses based on *COI* genes agree quite closely with the systematic arrangement of the genera based on the morphological characters.

The subfamily classification scheme suggested by *Abe & Smith (1991)* is confirmed by the results of the present analyses. The monotypic Corynidinae are always retrieved with
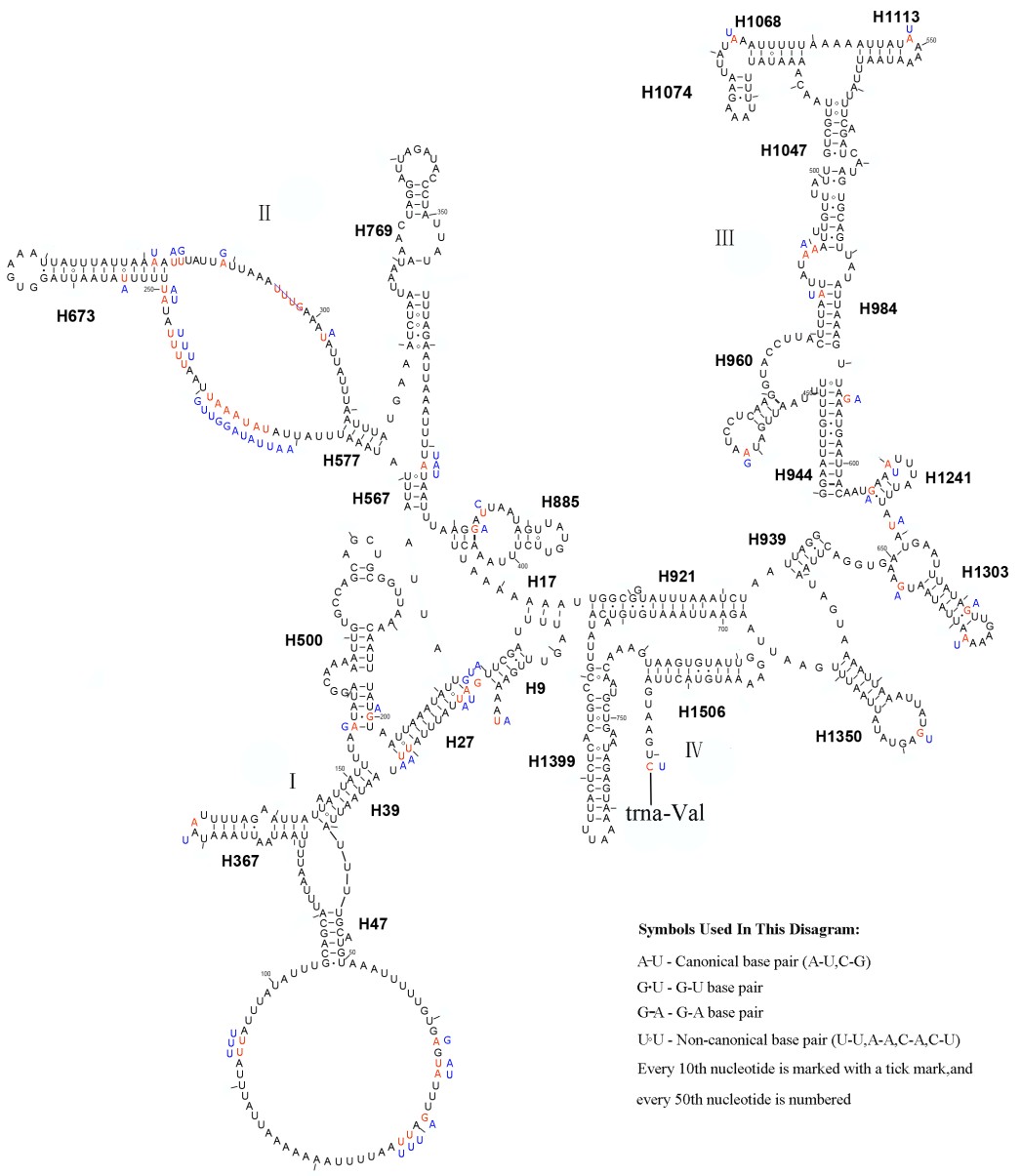

**Figure 10  The predicted secondary structures of *rrnS* of *L. sinicus* and *T. anthracinum*.** Tertiary interactions and base triples are connected by continuous lines. The numbering of helix follows *Gillespie et al. (2006)*. Roman numbers refer to domain names. Dashes indicate Watson-Crick base pairing and dots indicate G-U base pairing. The helical variation among cimbicid species are presented in boxes with red (*L. sinicus*) and blue (*T. anthracinum*) color.

strong support, and Abiinae + Cimbicinae is best-supported internal node in the present analyses of Cimbicidae, which was also supported by *Vilhelmsen (2019)*.

The clade consisting of ((((*Labriocimbex + Praia*) + *T. anthracinum*) + *Trichiosoma*) + *Leptocimbex*) + *Cimbex*, was highly supported in both trees (posterior probability >0.9511 and bootstrap support >87). *Labriocimbex* is always retrieved as monophyletic (two samples

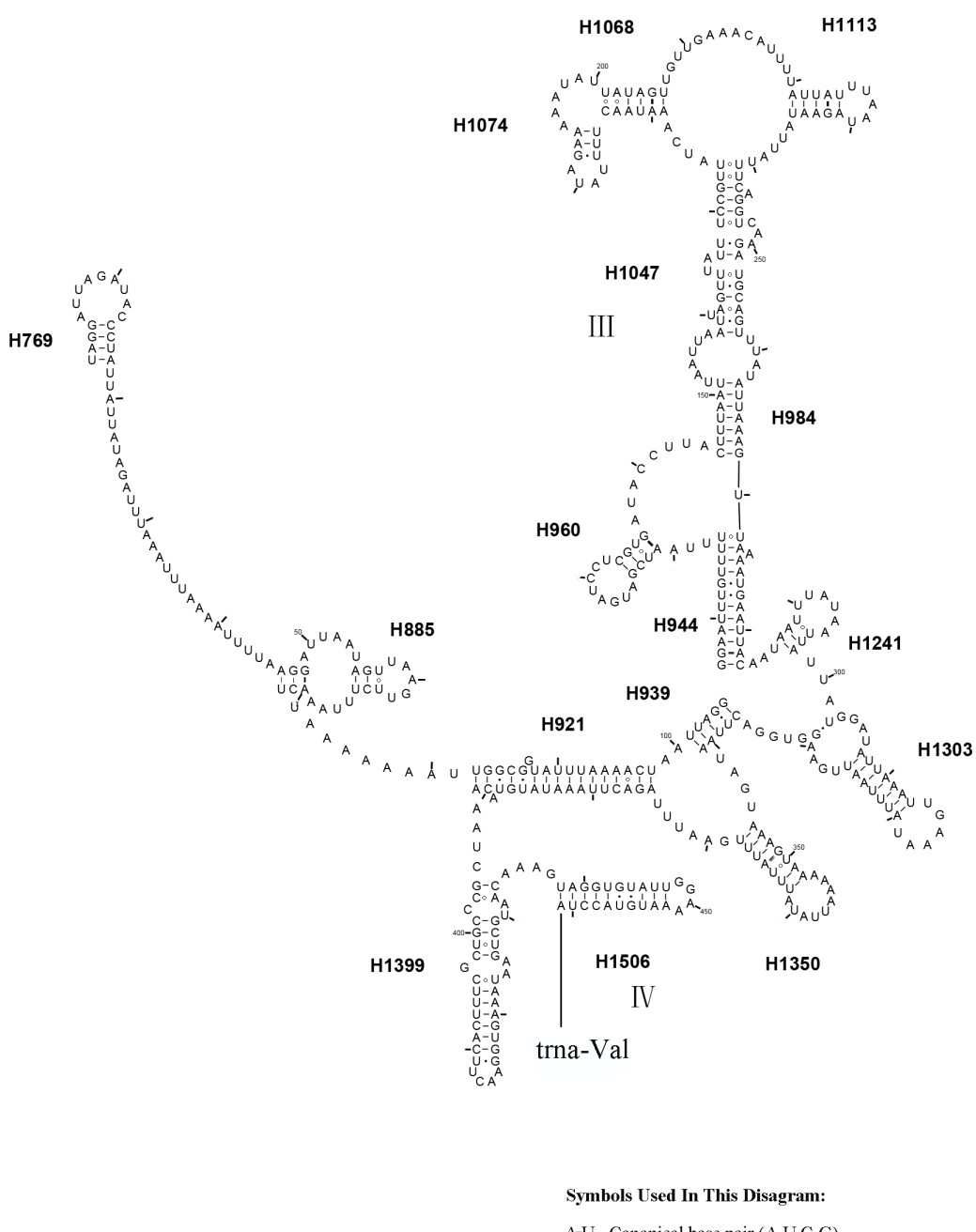

**Figure 11** *Corynis lateralis rrnS.* Predicted *rrnS* secondary structure in *C. lateralis.* The numbering of helix follows *Gillespie et al. (2006)*. Roman numbers refer to domain names.

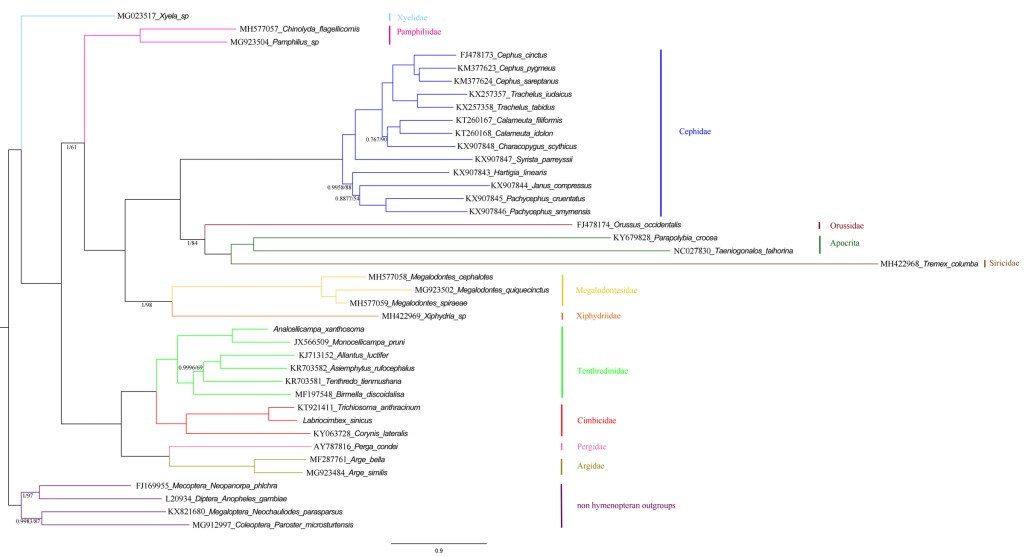

**Figure 12  Symphytan phylogenetic tree constructed with BI and ML approaches using a mitochondrial genome dataset including 15 individual genes (13 PCGs and two rRNAs).** Both analyses produced the same tree topology. Support values lower than 100% in the ML analysis and 1.0 in the BI analysis were shown.

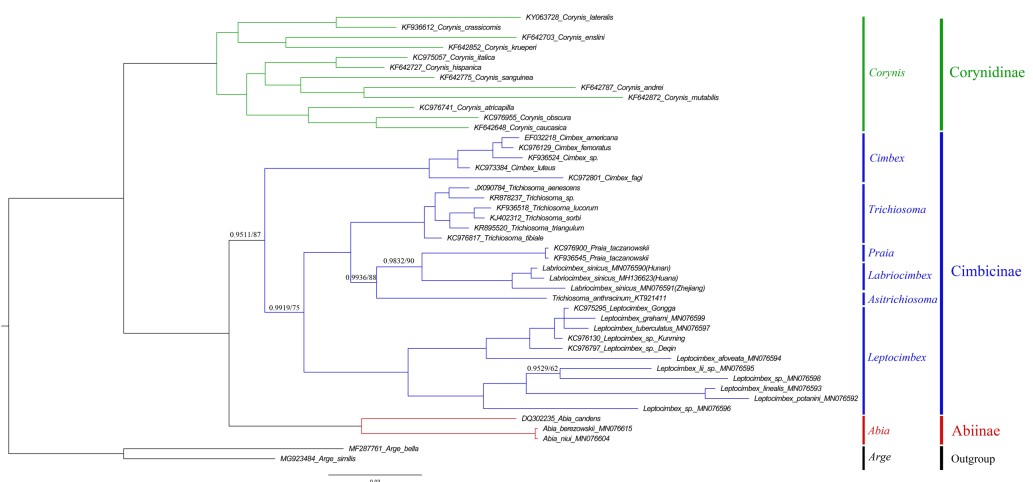

**Figure 13  Cimbicidae phylogenetic tree constructed with BI and ML approaches using the *COI* gene data.** Both analyses produced the same tree topology. Support values lower than 100% in the ML analysis and 1.0 in the BI analysis were shown.

were collected in Hunan, one was collected in Zhejiang), and it is the sister group of *Praia*. *Praia* + *Labriocimbex* is the sister group of *T. anthracinum*. *Malaise (1939)* placed *T. anthracinum* and other two species, *T. hymalayensis* and *T. sikkimensis*, in the subgenus *Asitrichiosoma* within *Trichiosoma*. According to our analyses (Fig. 13), *T. anthracinum* renders the genus *Trichiosoma* paraphyletic, and it is necessary to reestablish the genus *Asitrichiosoma*.

Additionally, we demonstrated that *COI* gene sequences can be used to solve phylogenetic relationships between genera of Cimbicidae. However, to reconstruct the generic phylogeny of Cimbicidae the further studies of the mitochondrial genomes and external morphology of more taxa of the family are needed.

## CONCLUSIONS

*Labriocimbex* gen. nov. was regarded similar to the genera *Trichiosoma* Leach and *Praia* Andre. Most of the characteristics of the new genus suggest placing it in the tribe Trichiosomini. The most important characteristics include: the labrum large with the basal breadth about half the breadth of clypeus, the jugum region in hind wing without crossvein, the clypeus very short and much broader than lower distance between eyes and not merging with supraclypeal area. Following characters help to distinguish this new genus and new species: the clypeus and labrum black; the clypeus broadly and shallowly emarginated; the labrum triangular and tapering toward apex, basal breadth about half the breadth of clypeus; the apical anal cell about 2 times as long as basal anal cell; the hind femora close to each other and without ventral denticle; the very large tarsal pulvilli; malar space 2.3 times the diameter of lateral ocellus; the inner margins of eyes parallel; head distinctly dilated behind eyes; the inner spur of hind tibia as long as apical breadth of tibia, apex blunt and membranous; the long and dense hairs covering head, thorax, base of abdomen and legs.

The complete mitochondrial genome of *L. sinicus* was obtained and was found to have a length of 15,405 bp and a typical set of 37 genes. The secondary structures of the 22 tRNAs and two rRNAs resemble those of Symphyta. In comparison with the structures of *T. anthracinum* and *C. lateralis*, some helices were highly variable in *rrnL* and *rrnS*.

The same cladograms were obtained using two different analytical methods, and our findings partly disagreed with traditional morphological classification. The tree topology confirmed that Cimbicidae is a member of the superfamily Tenthredinoidea and *Labriocimbex* gen. nov. is a member of Cimbicidae.

Within Cimbicidae, we have made several interesting discoveries, including a proposal to place the *Praia* into tribe Trichiosomini, and to promote the subgenus *Asitrichiosoma* to be a valid genus. Based on *COI* data, the phylogenetic position of *Labriocimbex* showed that it is the sister group of *Praia*, *Labriocimbex* + *Praia* is the sister group of *Asitrichiosoma anthracinum*, and *Labriocimbex* + *Praia* + *Asitrichiosoma* is the sister group of *Trichiosoma*. The positions of *Labriocimbex* and its close relatives remain to be decided in future studies. Here we suggest that *Labriocimbex* belongs to the tribe Trichiosomini of Cimbicinae based on adult morphology and molecular data.

## ACKNOWLEDGEMENTS

Thank Dr. Zejian Li of the Lishui Academy of Forestry, Zhejiang province, China, for collecting specimens from Mt. Tianmu. The members of Lab of Insect Systematics and Evolutionary Biology (LISEB) from Central South University of Forestry and Technology

are thanked for their contributions in laboratory work. We thank all the reviewers for their comments.

### Funding

This work was supported by the National Natural Science Foundations of China (No. 31672344; No. 31501885) and the Scientific Research Startup Funding for Talents Introduction, Jiangxi Normal University (2018-12019500). The funders had no role in study design, data collection and analysis, decision to publish, or preparation of the manuscript.

### Grant Disclosures

The following grant information was disclosed by the authors:
National Natural Science Foundations of China: 31672344, 31501885.
Scientific Research Startup Funding for Talents Introduction.
Jiangxi Normal University: 2018-12019500.

### Competing Interests

The authors declare there are no competing interests.

### Author Contributions

- Yuchen Yan conceived and designed the experiments, prepared figures and/or tables, authored or reviewed drafts of the paper, drafted the work or revised it critically for important content, approved the final draft of the manuscript submitted for review and publication.
- Gengyun Niu conceived and designed the experiments, prepared figures and/or tables, approved the final draft of the manuscript submitted for review and publication.
- Yaoyao Zhang analyzed the data, performed the experiments, prepared figures and/or tables, approved the final draft of the manuscript submitted for review and publication.
- Qianying Ren performed the experiments, analyzed the data, prepared figures and/or tables, approved the final draft of the manuscript submitted for review and publication.
- Shiyu Du analyzed the data, prepared figures and/or tables, approved the final draft of the manuscript submitted for review and publication.
- Bocheng Lan prepared figures and/or tables.
- Meicai Wei conceived and designed the experiments, drafted the work or revised it critically for important content, approved the final draft of the manuscript submitted for review and publication.

### DNA Deposition

The following information was supplied regarding the deposition of DNA sequences:
Labriocimbex sinica sequences are available at GenBank: MH136623 and SRA: SRR8270383.

Sequences and raw data are also available at Figshare: Yan, Yuchen; Niu, Gengyun; Zhang, Yaoyao; Ren, Qianying; Du, Shiyu; lan, Bocheng; et al. (2019): Complete mitochondrial genome sequence of Labriocimbex sinicus, a new genus and new species of Cimbicidae (Hymenoptera) from China. figshare. Dataset. https://doi.org/10.6084/m9.figshare.7339334.v1.

## Data Availability

The raw data is available at Figshare: DOI 10.6084/m9.figshare.7339334.v1.

Data is also available at NCBI Genbank accession numbers MN076590–MN076605, MH136623 and NCBI SRA accession number SRR8270383.

The holotype and some paratypes of the new species are deposited in the Asian Sawfly Collection, Nanchang, China (ASCN). The most remaining paratypes are deposited in the Insect Collection of Central South University of Forestry and Technology, Changsha, Hunan, China (CSCS). A few paratypes are kept in Lishui Academy of Forestry (LSAF). Specimen voucher numbers: CSCSHymM00003, CSCSHymM02146, CSCSHymM02107, CSCSHymM02110, CSCSHymM02114, CSCSHymM02119, CSCSHymM02120, CSCSHymM02100, CSCSHymM02112, CSCSHymM02108, CSCSHymM02102, CSCSHymM02103, CSCSHymM02109, CSCSHymM02115, CSCSHymM02117, CSCSHymM00393, CSCSHymM00009, CSCS13010_Lab001, CSCS13010_Lab002–033, CSCS13015_Lab034–061, CSCS13014_Lab062–123, CSCS11009_Lab124–210, CSCS05001_Lab211–249, CSCS1999001_Lab250, LSAF18029_Lab251–258, LSAF17053_Lab259, LSAF17054_Lab259–285, CSCS18006_Lab286–291, CSCS18007_Lab292–299, CSCS09019_Lab300.

## New Species Registration

The following information was supplied regarding the registration of a newly described species:

Publication LSID: urn:lsid:zoobank.org:pub:EE7F5193-78B2-42CE-87C1-B3FE947CB70F

Labriocimbex Yan & Wei, gen. nov. LSID: urn:lsid:zoobank.org:act:29EB6C0E-881D-46E2-AEF0-3BDF5992EC37

Labriocimbex sinicus Yan & Wei sp. nov. LSID: urn:lsid:zoobank.org:act:E1454ED2-5321-4D39-97C2-EC8957D034C1.

## Supplemental Information

Supplemental information for this article can be found online at http://dx.doi.org/10.7717/peerj.7853#supplemental-information.

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
