# Peer review of "Complete mitochondrial genome sequence of Labriocimbex sinicus, a new genus and new species of Cimbicidae (Hymenoptera) from China"

_PeerJ, doi:10.7717/peerj.7853_

## Round 0.1 · original submission · Major Revisions

Dear Dr. Yan,

Thank you for your submission to PeerJ.

I carefully read two reviewers' suggestions. I think their suggestions are very helpful to improve your manuscript.

I found the genus Labriocimbex is reported in the paper titled "The Investigation and Taxonomical Research of the Sawflies Species from Gansu Province" in 2010 (Li and Wu). The Chinese name of the paper is "甘肃叶蜂种类调查及分类研究Ⅲ. 锤角叶蜂科属种名录" . You should read this paper. If Labriocimbex sinica belong to this genus, you should rewrite your new species and compare the characters to Labriocimbex pilosus sp. nov. (I failed to find any charachters description in Labriocimbex pilosus). One author in your paper is the collector of this species.
If you thought "Labriocimbex pilosus or Labriocimbex " is invalid, I agree the reviewer's comment "While the authors are thus correct in asserting that their current submission will validate the names, they need to address explicitly the situation with the previous nomina nuda status; future readers might come across these previous publications and wonder if the current publication has priority, so it is better to explain the situation in full."

I did not think the phylogenetic relationship among Cimbicidae is helpful to support the new species or new genus. I think the dercription of mitochondrial gene arrangement and characters are enough.If you want to discuss the relationship among Cimbicidae, you should download all known mitochondrial genomes of Cimbicidae to further analyze.

My suggested changes and reviewer comments are shown below and on your article 'Overview' screen.

If you address these changes and resubmit, there's a good chance your article will be accepted (although this isn't guaranteed).

Jia-Yong

·

Basic reporting

see below

Experimental design

see below

Validity of the findings

see below

Additional comments

Review of Yan et al. (PeerJ#34672)
The submitted paper provides description of a new genus and species of a sawfly of the family Cimbicidae. In addition, the complete mitochondrial genome of the species is reported. The information presented is new and interesting and should be published. The descriptions of the new taxa and mitogenome are adequate; the morphological illustrations should be improved. Furthermore, the taxonomic and phylogenetic context of the new taxon is not properly established and discussed and needs to be improved.
Some more specific points:
- The authors are careful to point out (l. 119-122) that the electronic version of the paper, when published, constitutes the valid publication of the scientific name Labriocimbex sinica. This seems a bit superfluous, unless you realize that the name has appeared in print before: Labriocimbex in Li & Wu (2010) and Labriocimbex sinicus [sic!] Vilhelmsen (2019). In the case of the genus name, Labriocimbex, this means that it has been a nomen nudum for almost a decade. The two earlier mentions of the names (there might be more that I am not aware of) were not accompanied by a proper description, so they are not available in the sense of the ICZN code. While the authors are thus correct in asserting that their current submission will validate the names, they need to address explicitly the situation with the previous nomina nuda status; future readers might come across these previous publications and wonder if the current publication has priority, so it is better to explain the situation in full.
- As indicated above, the proper full name of the new taxon is Labriocimbex sinicus as genus names ending in –cimbex are regarded as masculine in grammatical gender and sinicus/a is an adjective, it needs to follow the gender of the genus name (see Article 31.2 of the ICZN code).
- The illustrations need to be improved. At least in the version I received for review, Figs 1 & 2 are of low resolution and with much highlight (e.g., Fig. 2A). The authors do not mention any attempts to disperse the light during photography; this can be simply done by placing a cylinder/cone of semitransparent paper or plastic around the specimen, and between it and the source(s) of light. I have a set of images of male and female of Labriocimbex which almost matches that of the authors, but of much better quality; one is published in Vilhelmsen (2019; fig. 3B).
- In addition to describing Labriocimbex sinicus as the type species of the genus, the authors transfer Trichiosoma zaraeoides Malaise, 1939 to the new genus as Labriocimbex zaraeoides (Malaise, 1939) comb. nov. (l. 340). No mention is made of what material they examined of this taxon (did they examine the holotype?), in contrast to the detailed information provided for the material of Labriocimbex sinicus. Also, they should provide illustrations of Labriocimbex zaraeoides so the reader can see simliarites in differences between the two taxa; there is no real justification provided for the comb. nov., it is just ‘similar to L. sinica’ (l. 343) and then some differences are mentioned.
- The authors state (l. 241) that ‘the new genus is similar to Pseudoclavellaria … and Trichiosoma’. There is no explicit statement at what characters indicate the similarity, but some differences are listed. Later, it is stated that Labriocimbex belongs to Trichiosomini (ls 518-19), and it is included in the key for this tribe (ls 250-266). There is no justification provided for this placement or the monophyly of Trichiosomini in the paper. Only three genera of Cimbicidae are sampled, two from Trichiosomini and one from Corynidinae (obs: not Coryninae!). Since no Cimbicinae outside Trichiosomini are included, the monophyly of the latter tribe is not tested. No explicit autapomorphies for the tribe is mentioned either. This contrasts with the results of Vilhelmsen (2019), which did not retrieve Trichiosomini, despite a much larger taxon sample (i.e., 100 cimbicid terminals, with all genera of Cimbicinae represented). The main reason for the authors to uphold Trichiosomini seems to be the unpublished thesis quoted as Deng (2000). Since this is apparently in Chinese and not published, it should not be included in the References; references are publications that can be checked by readers (this is why papers include a reference list!), this is evidently not the case for Deng (2000). If the authors want to cite information from this thesis, they should explain more about the taxon and character sampling of this paper, and discuss its results in comparison with those of Vilhelmsen (2019).
- The choice of taxa for the phylogenetic analyses of mitogenomes is not properly introduced until the beginning of the Discussion (ls. 494-509); this paragraph should be placed in the Materials & Methods section. Furthermore, there should be more justification for the choice of taxa included. Specifically, why so many Cephidae and why so few Apocrita? Both are essentially outgroups when trying to place the new taxon, which undoubtedly belongs in the Tenthredinoidea. The inclusion of 13 species of Cephidae is overkill as these have already been analyzed in the Korkmaz et al. (2018) which showed Cephidae to be monophyletic. 3-4 representatives from Cephidae should be more than enough for the present paper, as long as both Cephini and Hartigiini are represented. I suspect the rationale is to include all ‘symphytan’ mitogenomes published; however, the ‘Symphyta’ – Apocrita subdivision is completely arbitrary as the former ‘suborder’ is paraphyletic. The only consistent way of sampling mitogenomes would be to include all those produced for Hymenoptera, but this would be beyond scope of the paper.
- The discussion of the phylogenetic results (l. 510-521) is very cursorial and glosses over the substantial differences between the topologies of the other papers cited (in particular with that of Peters et al. (2017) and the rest), and between these and the current submission. For instance, all the papers cited (except for Vilhelmsen 2015, which did not include any carnivores) retrieved carnivorous Hymenoptera (Orussidae + Apocrita) in contrast with the present submission which has Siricidae sister to Apocrita, or even inside Apocrita. This is a highly unusual (and dubious) result and should be mentioned. In general, the language treatment of relationships in the paper is vague and imprecise, e.g., ‘grade basal to’ (l. 93), ‘connecting link’ (l. 524), ‘similar to’ (recurrent phrase). When you have a tree and want to discuss relationships, talk explicitly about sister groups. Why else produce a tree?
Additional comments are in the annotated pdf.
Best regards
Lars Vilhelmsen

Reviewer 2 ·

Basic reporting

This manuscript describes a new species from a new genus from Cimbicidae and reports its mitochondrial genome, presents comparative analyses of this mitogenome with the other known symphytan in particular cimbicid mitogenomes and a phylogenetic analysis of Symphyta based on the mitogenome sequences. The manuscript is clearly written.

Experimental design

There seems to be no methodological problem.

Validity of the findings

There are important results for the field and the findings are well constructed by authors. The findings have been discussed and supported by relevant literature sufficiently.

Additional comments

Here, the manuscript presents the characterisation of the mitochondrial genome of L.sinica for the first time. Then the authors aimed to comparison of various genomic features against other reported cimbicid species. The most interesting evidence is the presence of rearrangements in tRNA genes. The manuscript presents a newly constructed phylogenetic tree using available symphytan mitochondrial genome to validate the phylogenetic position of the species (L.sinica). Some minor changes or comments are listed below:
• Line 187: It would be better if the authors prefer presenting a supplementary table explaining the best partitioning schemes and associated models.
• Line 380: When looking Table 4, Leu, Ile, Phe, and Ser seem to be most frequently used amino acids rather than indicated by the authors.
• Lines 386-388: I found two codons as absent and nine codons as used once. In my opinion, two times or four times used codons are also acceptable as rare. The related part should be revised.
• Line 518-519: The authors have stated that Labriocimbex belongs to the tribe of Trichiosomini as well as Trichiosoma. However, it could not be suggested when looking through the trees. In the trees, there are only three cimbicid species from three genera representing two subfamilies. Therefore, without expanding the sampling belong to same genus as well as same tribus, we could not decide that this new genus belongs to the tribe of Trichiosomini.
• Line 538-545: It would be better if the authors rewrite the phylogeny related part of the conclusions section.
• The remaining minor corrections are shown on the manuscript.
In my opinion, the manuscript could be published in the journal PeerJ after major revision.

Annotated reviews are not available for download in order to protect the identity of reviewers who chose to remain anonymous.

---

## Round 0.2 · Major Revisions

Dear Dr. Yan, and Prof. Wei,

I found that the part of phylogenetic relationship is still need to revise.
I failed to find the mt genome of Labriocimbex sincius in GenBank or Supplyment. Please add this sequence in supplyment when you resubmit the revised paper.

I also suggest if you want to discuss the classfication of New Genus Labriocimbex, you can add the part of phylogenetic relationship using COX1 as DNA barcoding which you can use more samples sequence to discuss the valid of new genus.

Spaces should be added between number and unit symbol.

The paper needs to be read and edited by someone proficient in English.

With kind regards,
Jia-Yong Zhang
Academic Editor, PeerJ

·

Basic reporting

See General comments

Experimental design

See General comments

Validity of the findings

See General comments

Additional comments

This is the second version I see of this paper. There has been substantial improvements to some parts of the paper. The illustrations are now improved and illustrations of Labriocimbex zaraeoides are included, strengthening the case for its inclusion in the genus. The nomina nuda issues are addressed explicitly; however, the authors briefly discuss ‘differences’ between the specimen that was previously labelled Labriocimbex pilosus and L. sinicus, although they apparently do not want to recognize the former as a separate species; this could actually be construed as an argument for recognizing it as a separate species. It would be safer to state that the morphology of this species falls within the known variation of L. sinicus and leave it at that.
The authors have added a key to Holarctic genera of Cimbicidae. This is fine in principle; however, they include at least three genera that have previously been synonymized with other genera (i.e., Orientabia and Zaraea with Abia and Palaeocimbex with Cimbex). Furthermore, the results of the phylogenetic analyses of Vilhelmsen (2019) which included species in the synonymized genera did not support the recognition of these genera. No justification for the recognition of these genera are provide by the authors; it would be better to leave them out.
The phylogenetic part of the paper is still quite weak. While the phylogenetic analyses are performed saisfactorily, mistakes in both interpreting the results and correctly discuss them abound (see annotations in paper). It is stated (ls 616-617): ‘We demonstrated that mitochondrial genome sequences can be used to solve phylogenetic relationships at different taxonomic levels within Symphyta’. This is far from the case; the taxon sampling is much too poor to provide any confidence in results below the superfamily level, especially in comparison with other recent, much more comprehensive analyses (e.g., Malm & Nyman 2015). The poor taxon sampling might explain some bizarre results obtained (Megalodontes sister to Xiphydria; Tremex inside or sister to Apocrita). In their response to reviewer’s comments, the authors state that the sister of Apocrita is not within the scope of the paper; this is fair enough, but they still need to acknowledge the shortcomings of their analyses, which are substantial. ‘Strong nodal support’ is not in itself a guarantee that a clade in a phylogeny will be robust to further testing – especially not in a dataset like this where you have a comparatively large amount of data (mitogenomes) but a fairly small taxon sample. Unless there is strong character conflict, you are bound to get high support for most nodes, also for relationships that will disappear as soon as more taxa are included.

Reviewer 2 ·

Basic reporting

I am happy to see the revised version of the manuscript about the mitogenome features and taxonomic position of Labriocimbex sinicus. I I feel that the text improved but I have highlighted some modifications on the manuscript attached.

Experimental design

no comment

Validity of the findings

I have question regarding the presentation of the phylogenetic figures (Figures 12 and 13) and explanation of the recovered relationship within the text. I think the authors should reepresent the phylogentic relationship on a single figure, because two approaches give a similar topology. The authors also suggest that “The recovered trees supported a relationship consisting of Xyelidae + (Tenthredinoidea + ((Megalodontesidae + Pamphiliidae) + (Xiphydriidae + (Cephidae + (Orussidae + (Apocrita + Siricidae)))))) in the Hymenoptera” in lines 577-579, but when checking the trees, a different relationship was found in the trees: Xyelidae + (Tenthredinoidea + (Pamphiliidae + ((Megalodontesidae + Xiphydriidae) + (Cephidae + (Orussidae + (Siricidae + Apocrita)))))) in Hymenoptera.

Additional comments

no comment

Annotated reviews are not available for download in order to protect the identity of reviewers who chose to remain anonymous.

---

## Round 0.3 · Major Revisions

The paper requies a number of Major Revisions. Please consider carefully the comments of reviewer 1. (eg. "Palaeocimbex is apparently not included in the sample and the authors do not discuss it, but retain it in the genus key without further justification."  “The type species of Orientabia, Abia egregia, is also not included. Without the inclusion of the type species, how can the authors be sure that they have correctly delimited these genera?”…)

·

Basic reporting

see below

Experimental design

do

Validity of the findings

do

Additional comments

This is the third version of this paper I have seen. I have not gone through the text in detail again and will just comment on the main new addition to the paper, the CO1 dataset and analysis.
The authors want to make a case that the genera Orientabia, Zaraea and Palaeocimbex that have been previously synonymized with Abia and Cimbex, respectively, should be recognized as valid. To this end, they present a COI data set for 52 cimbicid terminals and two Arge spp. as outgroup. The sequences are primarily obtained from Genbank and the cimbicid sample contains a number of duplicates at the species level and many taxa unidentified to species. Palaeocimbex is apparently not included in the sample and the authors do not discuss it, but retain it in the genus key without further justification.

In the case of Abia, Orientabia and Zaraea, all are apparently included in the taxon sample, but in low numbers (three named species for Abia and Zaraea each and one Orientabia [seems to be missing in Table 2], as well as two Zaraea sp. which come out together in Fig. 13 and might be the same species). This is not an impressive taxon sample for Abiinae that comprises 60+ described species and barely allows for confirming the monophyly of Abia and Zaraea, while Orientabia is not tested. Curiously, only one taxon (Trichiosoma sorbi) from Liston et al. (2014) have been included despite this paper providing sequences for a number of additional taxa not in the present sample. This goes in particular for Abiinae; crucially Liston et al. included Abia sericea (Linné, 1767) and Abia fasciata (Linné, 1758), the type species of Abia and Zaraea, respectively. The type species of Orientabia, Abia egregia, is also not included. Without the inclusion of the type species, how can the authors be sure that they have correctly delimited these genera? Liston et al. (2014; fig. 1A) in their COI tree retrieved A. fasciata and A. sericea as closely related; their taxon sample for COI was similar in size (six species) to that of the current submission, but with little taxon overlap (Abia candens only).

The authors conclude (ls 691-692): ‘The clade of Orientabia, Zaraea and Abia, are recovered as monophyletic with high support for the both trees’. Assuming this means that they consider to have tested and confirmed the monophyly of all three genera, this is clearly not the case for Orientabia, which is only represented by one terminal; furthermore, having a one-gene data set with poor taxon sampling for the two other genera is hardly strong evidence either. Additionally, the results obtained do not seem to conform to that of Liston et al (2014) for Abiinae or for Abiinae and Cimbicinae when compared with Vilhelmsen (2019); this is not discussed at all, the authors are content to focus on their own data set as has been the case with previous versions of the submission. Similarly, it is stated (ls 636-638): ‘We found that the generic relationships of Cimbicidae revealed by the phylogenetic analyses based on COI genes agree quite closely with the systematic arrangement of the genera based on the morphological characters’. This glosses over considerable differences with results for Cimbicinae in Vilhelmsen (2019), e.g., the position of Praia and Cimbex.

Finally, the authors assert (ls 653-654) that ‘the three genera [i.e, Orientabia, Zaraea and Abia] can also be distinguished morphologically’. This is not surprising given that this was why they were defined in the first place, but that is hardly the point. Rather, the question is whether the character combinations used to define the different genera are delimiting natural (i.e., monophyletic) groups or are just random combinations; if the latter is the case, this can result in a proliferation of genera that are not of any use to anyone beside those who take pleasure in defining them. The conclusions of the authors in this regard runs contrary to the taxonomic treatments of, e.g. Taeger (1998) and Hara & Shinohara (2017; they synonymized Orientabia, why is this paper not cited?), as well as the phylogenetic treatments of Liston et al. (2014) and Vilhelmsen (2019), who all argued for placing all the Abiinae species they examined in Abia. The comparatively weak evidence presented in the current submission does not outweigh this.

Reviewer 2 ·

Basic reporting

I am happy to see the revised version of the manuscript about the mitogenome features and taxonomic position of Labriocimbex sinicus. I feel that the text improved but I have highlighted some minor linguistic comments or modifications on the manuscript attached. Also, the writing and presentation of the newly added part on the phylogenetic relationship of Cimbicidae using the COI barcode region need to rewritten to increase the clarity and flow of the manuscript.

Experimental design

no comment

Validity of the findings

no comment

Annotated reviews are not available for download in order to protect the identity of reviewers who chose to remain anonymous.

---

## Round 0.4 · Minor Revisions

Please correct the paper according to the final, minor, reviewers' suggestions.

I also suggest the authors should check the format of references.

·

Basic reporting

see below

Experimental design

see below

Validity of the findings

see below

Additional comments

I think the current version is better than the previous one in the sense that the parts considering the generic classification of Abiinae has been remove and the key adjusted (you might want to call it 'Key to extant Holarctic genera of Cimbicinae' as it only properly deals with this subfamily); the genus names are still retained in Fig. 13 though, with no justification for using Orientabia and Zaraea. Also, you should be aware that Orientabia magna Takeuchi, 1939 is now considered a junior synonym of Abia relativa Rohwer, 1910; see Hara & Shinohara (2017).
The taxon sample for the mitogenomic analyses are still highly inadequate (especially with regard to Apocrita) and some of the results are weird: Siricidae as sister to Apocrita, Megalodontesidae as sister to Xiphydriidae; the latter is completely ignored, and neither is likely to hold up to further scrutiny.
The text should be checked for the use of 'nomina nuda'; this is plural, the singular form is 'nomen nudum'; you cannot say 'nomina nudum' (l. 279), and the plural should only be used when you discuss two or more names (l. 443, but not l. 99)

Reviewer 2 ·

Basic reporting

I am happy to see fourth round of the manuscript about the mitogenome features and taxonomic position of Labriocimbex sinicus. I feel that the text mostly improved by the authors. In my suggestion, the manuscript is now accepted in PeerJ.

Experimental design

no comment

Validity of the findings

no comment

---

## Round 0.5 · accepted · Accept

I am writing to inform you that your manuscript has been Accepted for publication. Congratulations!